# RLP: Reinforcement as a Pretraining Objective

**Ali Hatamizadeh**[†1], **Syeda Nahida Akter**[†2,*], **Shrimai Prabhumoye**[†1,3], **Jan Kautz**[1],
**Mostofa Patwary**[1], **Mohammad Shoeybi**[1], **Bryan Catanzaro**[1], **Yejin Choi**[1,4]
NVIDIA[1], Carnegie Mellon University[2], Boston University[3], Stanford University[4]
ahatamizadeh@nvidia.com, sprabhumoye@nvidia.com

## Abstract

The dominant paradigm for training large reasoning models starts with pre-training using next-token prediction loss on vast amounts of data. Reinforcement learning, while powerful in scaling reasoning, is introduced only as the very last phase of post-training, preceded by supervised fine-tuning. While dominant, is this an optimal way of training? In this paper, we present RLP, an information-driven reinforcement pretraining objective, that brings the core spirit of reinforcement learning—exploration—to the last phase of pretraining. The key idea is to treat *chain-of-thought* as an exploratory action, with rewards computed based on the *information gain* it provides for predicting future tokens. This training objective essentially encourages the model to think for itself before predicting what comes next, thus teaching an independent thinking behavior earlier in the pretraining. More concretely, the reward signal measures the increase in log-likelihood of the next token when conditioning on both context and a sampled reasoning chain, compared to conditioning on context alone. This approach yields a verifier-free dense reward signal, allowing for efficient training for the full document stream during pretraining. Specifically, RLP reframes reinforcement learning for reasoning as a pretraining objective on ordinary text, bridging the gap between next-token prediction and the emergence of useful chain-of-thought reasoning. Pretraining with RLP on QWEN3-1.7B-BASE lifts the overall average across an eight-benchmark math-and-science suite by 19%. With identical post-training, the gains compound, with the largest improvements on reasoning-heavy tasks such as AIME25 and MMLU-Pro. Applying RLP to the hybrid NEMOTRON-NANO-12B-V2 increases the overall average from 42.81% to 61.32% and raises the average on scientific reasoning by 23%, demonstrating scalability across architectures and model sizes.

## 1 Introduction

Large Language Models (LLMs) pretrained with next-token prediction loss have demonstrated broad utility, but this objective does not explicitly encourage long-range reasoning or integration with world knowledge. Consequently, state-of-the-art models (Guo et al., 2025; Yang et al., 2025) rely on post-training objectives such as supervised fine-tuning (SFT) and reinforcement learning with human or verified feedback (RLHF, RLAIF, RLVR) (Ouyang et al., 2022; Lambert et al., 2024) to induce complex reasoning abilities. In contrast, human comprehension is not a linear token-by-token process, but rather a parallel integration of input with prior knowledge (Baumgaertner et al., 2002; Hagoort et al., 2004; Metzner et al., 2015). Current pretraining lacks such mechanisms, limiting the model's ability to reason and ground language in world knowledge during learning.

To fill this gap, we propose **R**einforcement **L**earning **P**re-training (RLP) which treats Chain-of-Thought (CoT) generation as an explicit action taken before predicting each next token. As shown in Fig.1, the model first samples an internal thought, then predicts the observed token from the same context augmented with that thought. The training signal is the increase in log-likelihood of the observed token when the thought is present compared to a no-think baseline. This yields a verifier-free and dense reward that assigns position-wise credit wherever thinking improves prediction. Because

---

*Work done during internship at NVIDIA
† Equal contribution

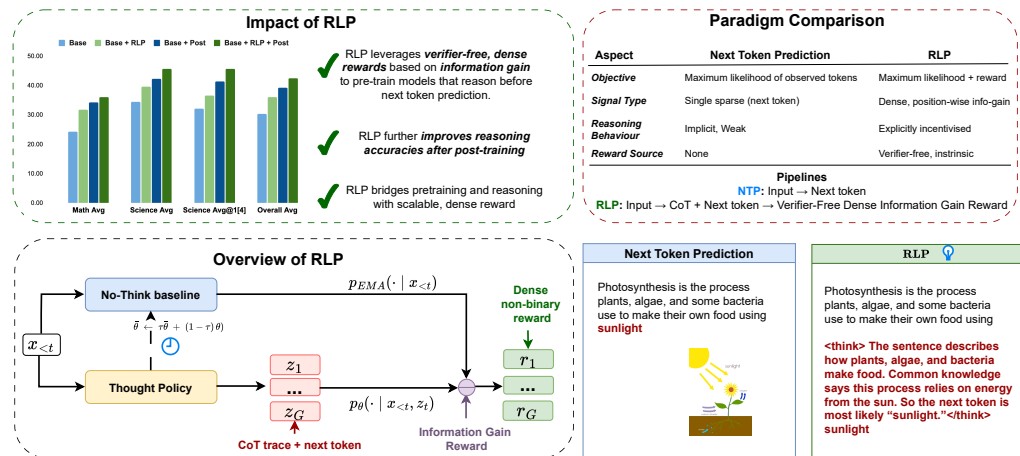

Figure 1: **Visualization of the RLP framework.** A chain-of-thought is sampled *before* next-token prediction. Rewards are computed by contrasting the predictor conditioned on the CoT with a *No-think* EMA baseline, yielding a verifier-free, dense signal. We list the advantages of RLP over the traditional pretraining objective (top right) and show the impact after end-to-end training (top left).

the signal is defined for ordinary text with teacher forcing, RLP reframes reinforcement learning for reasoning as reinforcement pretraining on the same streams used for maximum likelihood.

Unlike post-training with verifiable rewards, which requires task-specific checkers or curated solutions, RLP is verifier-free: the signal is computed directly from log-evidence under the model and a baseline, allowing uniform application to domain agnostic web-scale text. Compared to reinforcement pretraining via prefix-matching rewards (RPT) (Dong et al., 2025), which uses sparse binary reward and often relies on proxy-model filtering of "easy" tokens, RLP provides a continuous improvement signal at every position and trains on the full documents. This eliminates the need to preselect high-entropy tokens or couple training to a separate small model. Prior RPT demonstrations also depend on distilled checkpoints with strong prior reasoning ability, which clouds whether the method helps base models. RLP is designed to shape thinking in base models by rewarding only those thoughts that measurably help next-token prediction.

This work makes the following key contributions: We introduce **RLP, a verifier-free information-gain objective** that augments next-token prediction by rewarding thoughts in proportion to their predictive utility. We develop a **practical and stable training algorithm** that interleaves reinforcement updates with standard likelihood training via group-relative advantages, a clipped surrogate for thought tokens, and a slowly updated Exponential Moving Average (EMA) baseline. We provide **theoretical guarantees** linking expected reward to reductions in cross-entropy and to a computable lower bound, ensuring both interpretability and tractability. We conduct comprehensive experiments showing that RLP outperforms strong baselines, remains robust after strong post-training, generalizes across diverse corpora, and scales effectively to larger model sizes and hybrid architectures—establishing it as a broadly applicable reinforcement pretraining objective.

Our empirical validation is comprehensive, assessing the efficacy of RLP along four key axes. First, we evaluate its performance relative to traditional next-token prediction baselines. On the QWEN3-1.7B-BASE model, RLP outperforms continuous pretraining by $+17\%$ and RPT by nearly $+4\%$. We show the advantage persists even when the baseline uses $35\times$ **more data** to match FLOPs, confirming the gains arise from methodology rather than compute. Second, we demonstrate the robustness of these improvements, showing they are not transient. As shown in Fig.1, when subjected to an identical, strong post-training regimen, the foundational advantages of RLP **compound**, allowing our final model to surpass its conventionally trained counterparts by a significant 7–8% margin. Third, unlike methods requiring narrow, curated datasets, RLP successfully extracts a powerful reasoning signal from diverse, general-purpose web corpora–establishing its **versatility across data domains** (Table 4). Finally, we confirm its scalability and architecture-agnostic power. When applied to a 12B hybrid Mamba-Transformer (NEMOTRON-NANO-12B-V2), RLP achieves a staggering

**35% relative improvement** over a heavily trained baseline while using just 0.125% of the data—a testament to its remarkable data efficiency and broad applicability across LLM families and sizes.

## 2 METHODOLOGY

We introduce RLP, a pretraining-time procedure that explicitly induces reasoning. As illustrated in Fig. 1, RLP inserts a short Chain-of-Thought (CoT) *before* next-token prediction and measures how much that thought improves the model's log-probability of the observed token relative to a no-think baseline. This improvement, which is a log-likelihood ratio, is a verifier-free, dense reward available at every position in ordinary text corpora. By valuing thoughts in proportion to their predictive benefit, RLP turns reinforcement *pretraining* into learning to think on the same data used for standard next-token training.

**Parameterization and roles.**    We separate the components for clarity:

- **Thought policy / predictor** $\pi_\theta(c_t \mid x_{<t})$ and $p_\theta(x_t \mid x_{<t}, c_t)$ share *exactly the same* network and parameters $\theta$. The network first samples a CoT $c_t$ and then, conditioned on the concatenated prefix $(x_{<t}, c_t)$, scores the next token $x_t$.

- **No-think baseline** $\bar{p}_\phi(x_t \mid x_{<t})$ (parameters $\phi$) is an EMA teacher of the current network used to score the same token without any CoT channel.

Thus, there is a single model that both *generates* the thought and *predicts* the next token given that thought; the EMA teacher provides the no-think counterfactual.

**Classical next-token objective.**    Given a text sequence $x = (x_0, \ldots, x_T)$ and position $t$, the standard next-token objective for a predictor $q_\eta$ is

$$\mathcal{L}_{\mathrm{NTP}}(\eta) := \mathbb{E}_{(x_{<t}, x_t) \sim \mathcal{D}}\big[\log q_\eta(x_t \mid x_{<t})\big]. \tag{1}$$

For distributions $p$ and $q$ on the next token, we define Cross-entropy (CE) as

$$\mathrm{CE}(p, q) \stackrel{\mathrm{def}}{=} \mathbb{E}_{x \sim p}\big[-\log q(x)\big]. \tag{2}$$

Using $p^*(\cdot \mid x_{<t})$ for the data distribution over $x_t$, maximizing equation 1 is equivalent to minimizing $\mathbb{E}_{x_{<t} \sim \mathcal{D}}\big[\mathrm{CE}\big(p^*, q_\eta(\cdot \mid x_{<t})\big)\big]$. *We include equation 1 only for context as our training **does not** include a standard NTP loss term.* Instead, RLP optimizes an information-gain objective defined below and updates parameters *only through the tokens of the sampled thoughts*.

### 2.1 REASONING AS AN ACTION

RLP augments next-token prediction with a sampled thought. At each position $t$, the policy draws a latent CoT random variable

$$z_t \sim \pi_\theta(\cdot \mid x_{<t}),$$

and we write $c_t$ for its realization. The network then predicts $x_t$ with the *reasoned* scorer $p_\theta(\cdot \mid x_{<t}, c_t)$. As a no-think counterfactual we use $\bar{p}_\phi(\cdot \mid x_{<t})$, the EMA teacher queried on the same context without providing the CoT.

**EMA teacher instantiation and schedule.**    We instantiate the EMA teacher to match the current model on the *first* batch ($\phi \leftarrow \theta$), and thereafter update it *after* each optimizer step via

$$\phi \leftarrow \tau \phi + (1 - \tau)\theta, \qquad \tau = 0.999.$$

This choice makes $\bar{p}_\phi$ a *moving counterfactual* that is (i) *current* enough to provide informative comparisons and (ii) *intentionally lagged* to mitigate reward hacking. If the baseline were frozen, the counterfactual would drift too far from the evolving model; if it tracked the model without lag, the log-likelihood ratio would collapse toward zero and invite degenerate strategies. The post-update averaging yields a one-step-lagged, smoothed teacher that stabilizes training.

## 2.2 Information-gain reward

With teacher forcing on the next token, define the reasoned and baseline log-evidence

$$S_{\text{pred}}(c_t) := \log p_\theta\big(x_t \mid x_{<t}, c_t\big), \tag{3}$$

$$S_{\text{EMA}} := \log \bar{p}_\phi\big(x_t \mid x_{<t}\big). \tag{4}$$

The *information-gain* reward is the log-likelihood ratio

$$r(c_t) := S_{\text{pred}}(c_t) - S_{\text{EMA}}, \tag{5}$$

which compares the reasoned scorer with a no-think baseline on the observed next token. Rewards are computed under teacher forcing for each $t$. When updating the policy, we *treat $r(c_t)$ as a constant with respect to $\theta$* (no backpropagation through $p_\theta$ or $\bar{p}_\phi$); see §2.4.

## 2.3 Expected improvement identity

**Proposition 1** (CE reduction). *For any fixed $(x_{<t}, c_t)$,*

$$\mathop{\mathbb{E}}_{x_t \sim p^*}[r(c_t)] = \text{CE}\big(p^*, \bar{p}_\phi(\cdot \mid x_{<t})\big) - \text{CE}\big(p^*, p_\theta(\cdot \mid x_{<t}, c_t)\big).$$

where $p^*(\cdot \mid x_{<t})$ is the data distribution over $x_t$. Maximizing the expected reward therefore maximizes the predictive usefulness of the thought for the next token.

**Proposition 2** (Lower bound via marginalization over thoughts). *Let $\pi_\theta(z_t \mid x_{<t})$ be the distribution over CoTs and define the collapsed predictor*

$$\tilde{p}_\theta(x \mid x_{<t}) = \mathop{\mathbb{E}}_{z_t \sim \pi_\theta(\cdot \mid x_{<t})}\big[p_\theta(x \mid x_{<t}, z_t)\big].$$

*Then for any realized $x_t$,*

$$\mathop{\mathbb{E}}_{c_t \sim \pi_\theta}\big[S_{pred}(c_t)\big] \leq \log \tilde{p}_\theta(x_t \mid x_{<t}), \quad \text{and} \quad J(\theta) = \mathbb{E}[r(c_t)] \leq \mathbb{E}\left[\log \frac{\tilde{p}_\theta(x_t \mid x_{<t})}{\bar{p}_\phi(x_t \mid x_{<t})}\right].$$

The CoT-conditioned objective is thus a computable lower bound on the improvement one would obtain after marginalizing thoughts. Refer to §8.1 of the appendix for the proofs of the propositions.

## 2.4 RLP objective and optimization

RLP optimizes the thought policy to produce thoughts that *increase* predictive evidence. Our training *does not* include the standard next-token loss in equation 1. Instead, we optimize only the information-gain objective

$$\max_\theta \; J(\theta) = \mathop{\mathbb{E}}_{x_{<t} \sim \mathcal{D}} \mathop{\mathbb{E}}_{c_t \sim \pi_\theta(\cdot \mid x_{<t})}\big[\, r(c_t) \,\big], \tag{6}$$

or, equivalently, we *minimize* the negative information-gain loss $\mathcal{L}_{\text{IG}}(\theta) = -J(\theta)$. Gradients are applied only to the *thought tokens*; $r(c_t)$ is treated as a constant (no backpropagation through $p_\theta$ or $\bar{p}_\phi$) .

**Group-relative baseline (inclusive mean with correction).** To reduce variance, for each context we sample $G \geq 2$ thoughts $\{c_t^{(i)}\}_{i=1}^G$ and use a corrected inclusive mean baseline. Let

$$\bar{r} = \frac{1}{G} \sum_{j=1}^{G} r(c_t^{(j)}).$$

We define the advantages

$$A^{(i)} := \frac{G}{G-1}\Big(r(c_t^{(i)}) - \bar{r}\Big), \qquad \text{with no gradient propagated through } \bar{r}. \tag{7}$$

This multiplicative factor removes the $\big(1 - \frac{1}{G}\big)$ shrinkage inherent to the inclusive mean, yielding an unbiased estimator with low variance.

---

**Algorithm 1** RLP for next-token prediction with information gain

---

1: **Inputs:** dataset $\mathcal{D}$, group size $G \geq 2$, clipping $(\epsilon_\ell, \epsilon_h)$, EMA decay $\tau \in (0, 1)$, learning rate $\eta$.
2: **Model:** a single network with parameters $\theta$ used both as (i) thought policy $\pi_\theta$ and (ii) reasoned predictor $p_\theta$; EMA baseline $\bar{p}_\phi$.
3: **Initialization:** mark $\phi$ as uninitialized.
4: **while** training **do**
5:      Set the behavior snapshot $\theta_{\text{old}} \leftarrow \theta$.              ▷ used for the current sampling pass
6:      Sample minibatch $\{(x_{<t}^{(b)}, x_t^{(b)})\}_{b=1}^{B} \sim \mathcal{D}$.
7:      For each $b$, sample $G$ thoughts $c_t^{(b,i)} \sim \pi_{\theta_{\text{old}}}(\cdot \mid x_{<t}^{(b)})$ with $|c_t^{(b,i)}| \geq 1$.
8:      **if** $\phi$ is uninitialized **then**
9:          $\phi \leftarrow \theta$                      ▷ lazy init of EMA teacher
10:      Compute baseline log-evidence (teacher forcing, no grad) $S_{\text{EMA}}^{(b)}$ as per equation 3.
11:      Compute reasoned log-evidence $S_{\text{pred}}^{(b,i)}$ and rewards $r^{(b,i)}$ as per equation 3 and equation 5.
12:      Group baseline $\bar{r}^{(b)}$ and $A^{(b,i)}$ (inclusive mean with correction; sg is stop-grad) as per equation 7.
13:      Per-token importance ratios and clipped surrogate for $\ell_u^{(b,i)}$ with prefix $\text{prefix}_u^{(b,i)}$:
$$\rho_u^{(b,i)} = \exp\Big( \log \pi_\theta(\ell_u^{(b,i)} \mid \text{prefix}_u^{(b,i)}) - \log \pi_{\theta_{\text{old}}}(\ell_u^{(b,i)} \mid \text{prefix}_u^{(b,i)}) \Big).$$
$$L_{\text{clip}}^{(b,i)} = -\frac{1}{|c_t^{(b,i)}|} \sum_u \min\Big( \rho_u^{(b,i)} \, \text{sg}\big(A^{(b,i)}\big), \; \text{clip}(\rho_u^{(b,i)}; 1 - \epsilon_\ell, 1 + \epsilon_h) \, \text{sg}\big(A^{(b,i)}\big) \Big).$$
14:      Policy update on thought tokens:
$$\mathcal{L}(\theta) = \frac{1}{BG} \sum_{b=1}^{B} \sum_{i=1}^{G} L_{\text{clip}}^{(b,i)}, \quad \theta \leftarrow \theta - \eta \nabla_\theta \mathcal{L}(\theta).$$
15:      EMA update of baseline: $\phi \leftarrow \tau \phi + (1 - \tau) \theta$.
16: **Output:** trained policy/predictor (shared $\theta$) and EMA baseline $\phi$.

---

**Per-token importance ratios and clipped surrogate.** We update the log-probability of the *thought* tokens with a clipped surrogate. Let $\ell_u^{(i)}$ be the $u$-th token in $c_t^{(i)}$ and $\text{prefix}_u^{(i)} = (x_{<t}, \ell_{1:u-1}^{(i)})$. With behavior parameters $\theta_{\text{old}}$ used to sample the thoughts, define the per-token importance ratio

$$\rho_u^{(i)} = \exp\Big( \log \pi_\theta(\ell_u^{(i)} \mid \text{prefix}_u^{(i)}) - \log \pi_{\theta_{\text{old}}}(\ell_u^{(i)} \mid \text{prefix}_u^{(i)}) \Big).$$

We write $\text{clip}(\rho; 1 - \epsilon_\ell, 1 + \epsilon_h)$ for elementwise clipping and denote stop-gradient by $\text{sg}(\cdot)$. The surrogate loss is

$$\mathcal{L}_{\text{clip}}(\theta) \;=\; -\mathbb{E}\left[ \frac{1}{|c_t^{(i)}|} \sum_u \min\Big( \rho_u^{(i)} \, \text{sg}(A^{(i)}), \; \text{clip}(\rho_u^{(i)}; 1 - \epsilon_\ell, 1 + \epsilon_h) \, \text{sg}(A^{(i)}) \Big) \right]. \tag{8}$$

## 2.5 REWARD PROPERTIES AND GUARANTEES

**Does thinking actually help?** The reward $r(c_t)$ is positive exactly when the model that used the sampled thought assigns higher probability to the observed next token than the EMA baseline that did not think. In expectation over the data distribution, this equals the reduction in cross-entropy between the reasoned scorer and the no-think baseline (Prop. 1).

**Positionwise credit at every step.** Since the task is next-token prediction, the reward is computed independently at each position $t$ as

$$r(c_t) \;=\; \log p_\theta(x_t \mid x_{<t}, c_t) \;-\; \log \bar{p}_\phi(x_t \mid x_{<t}).$$

Credit is attached exactly where the thought changes predictive probability, yielding one scalar per token and removing the need for a learned value function or any external verifier.

**Putting it all together.** Algorithm 1 composes the above pieces into a single training loop. Specifically, multiple thoughts are sampled per position and information-gain rewards are computed against a moving EMA counterfactual. Group-relative advantages are formed and the shared network is updated *only* on the thought tokens via the clipped surrogate in equation 8. In this case, the improvements originate from learning to generate thoughts that systematically raise predictive evidence.

| Benchmark | $\mathcal{M}_{\text{base}}$ | $\mathcal{M}_{\text{CPT}}$ | $\mathcal{M}_{\textbf{RLP}}$ | $\mathcal{M}_{\text{base}}$ **+Post** | $\mathcal{M}_{\text{CPT}}$ **+Post** | $\mathcal{M}_{\textbf{RLP}}$ **+Post** |
|---|---|---|---|---|---|---|
| AIME25 | 2.25 | 3.96 | **5.02** | 5.32 | 5.89 | **7.05** |
| MATH500 | 48.45 | 57.52 | **58.48** | 61.92 | 62.70 | **64.30** |
| GSM8K | 54.16 | 72.85 | **74.48** | 78.22 | 78.70 | **80.50** |
| AMC23 | 25.94 | 31.25 | **31.25** | 35.00 | 34.38 | **36.50** |
| Minerva | 15.30 | 19.03 | **21.19** | 25.30 | 26.10 | **27.80** |
| MMLU | 50.08 | 41.95 | **56.14** | 58.36 | 59.00 | **61.50** |
| MMLU@1[4] | 44.85 | 40.00 | **52.18** | 56.00 | 58.53 | **61.00** |
| MMLU-Pro | 28.17 | 27.81 | **34.62** | 37.85 | 39.92 | **42.40** |
| MMLU-Pro@1[4] | 23.95 | 24.61 | **30.80** | 36.53 | 38.49 | **41.30** |
| GPQA | 25.25 | 26.26 | **28.28** | 30.93 | 29.27 | **33.33** |
| GPQA@1[4] | 27.52 | 24.75 | **27.02** | 31.52 | 30.01 | **34.97** |
| Math Avg | 24.35 | 30.77 | **31.74** | 34.29 | 34.63 | **36.03** |
| Science Avg | 34.50 | 32.01 | **39.68** | 42.38 | 42.73 | **45.74** |
| Science Avg@1[4] | 32.11 | 29.79 | **36.67** | 41.35 | 42.34 | **45.76** |
| **Overall** | 30.32 | 30.85 | **36.03** | 39.34 | 39.90 | **42.51** |

Table 1: Quantitative benchmarks for Qwen3-1.7B-Base, showing the impact of RLP. Shaded columns indicate RLP variants; "Post" indicates SFT + RLVR post-training.

## 3 EXPERIMENTAL SETUP

We experiment with QWEN3-1.7B-BASE (Yang et al., 2025) and then scale our experiments to a larger NEMOTRON-NANO-12B-V2 (Nano, 2025) model.[1]

**RLP.** We apply RLP on a diverse set of datasets across two settings: (i) *SFT-style reasoning corpora*, including a math-centric set (OmniMath (Gao et al., 2024)) and mixed math + general-reasoning sets (OpenThoughts (Guha et al., 2025), Nemotron-Crossthink (Akter et al., 2025)); and (ii) *general-purpose pretraining corpora*, covering academic papers (ACAD), math textbooks (Math-Text), and open-ended web pages QA pairs from Common Crawl (Web-Crawl)(Nano, 2025). We train with RLP for 1B input tokens using general pretraining corpora ($\mathcal{D}_{\text{PT}}$) to evaluate its effect in an end-to-end LLM pretraining pipeline. We denote this model as $\mathcal{M}_{\text{RLP}}$. Note that theoretically RLP can be applied for every token in a document but in our experiments we randomly select one token per document. Hence, the number of tokens for which the reward is applied is far less than 1B.

**Continuous Pretraining.** To ensure compute equivalent comparison with $\mathcal{M}_{\text{RLP}}$, we do continuous pretraining on the base model denoted by $\mathcal{M}_{\text{base}}$ with the same tokens used in RLP. We denote this model as $\mathcal{M}_{\text{CPT}}$. This serves as an additional baseline for our experiments.

**Post-Training.** All models undergo a SFT stage on OpenThoughts data (Guha et al., 2025). To further enhance, we apply Reinforcement Learning with Verifier Rewards (RLVR) using MATH dataset (Hendrycks et al., 2021b). This two-stage post-training pipeline provides an evaluation framework to verify that gains from RLP persist under strong alignment, while also revealing how much additional improvement can be achieved through subsequent post-training. For consistency, all models are trained with identical SFT and RLVR receipes, ensuring that any observed differences in downstream accuracies can be attributed to the pretraining condition ($\mathcal{M}_{\text{base}}$ vs $\mathcal{M}_{\text{CPT}}$ vs $\mathcal{M}_{\text{RLP}}$).

### 3.1 EVALUATION METRICS

We conduct a thorough benchmark assessment using a series of tasks using NeMo-Skills[2].

---

[1]Details about hyper-parameters for each of the below phases and the prompt used for RLP can be found in Appendix 10.

[2]`https://github.com/NVIDIA/NeMo-Skills`

| Benchmark | $\mathcal{M}_{\text{base}}$ | $\mathcal{M}_{\text{RLP}}$ | $\mathcal{M}_{\text{base}}$ +Post | $\mathcal{M}_{\text{RLP}}$ +Post |
|---|---|---|---|---|
| MATH500 | 79.95 | **78.68** | 83.47 | **87.05** |
| GSM8K | 72.31 | **85.98** | 94.22 | **94.90** |
| AMC23 | 70.63 | **57.19** | 62.19 | **75.00** |
| Minerva | 22.61 | **39.48** | 40.76 | **42.78** |
| MMLU | 54.12 | **78.76** | 73.55 | **78.17** |
| MMLU@1[4] | 48.01 | **79.48** | 75.23 | **77.90** |
| MMLU-Pro | 24.16 | **53.13** | 61.78 | **67.38** |
| MMLU-Pro@1[4] | 27.13 | **55.76** | 73.21 | **66.96** |
| GPQA | 25.25 | **39.90** | 41.41 | **48.00** |
| GPQA@1[4] | 22.47 | **48.86** | 52.15 | **49.62** |
| Math Avg | 61.38 | **65.33** | 70.16 | **74.93** |
| Science Avg | 34.51 | **57.26** | 58.91 | **64.52** |
| Science Avg@1[4] | 32.54 | **61.37** | 66.86 | **64.83** |
| **Overall** | 42.81 | **61.32** | 65.31 | **68.09** |

Table 2: Quantitative benchmarks for NEMOTRON-NANO-12B-V2, showing the impact of RLP. Shaded columns indicate RLP variants; "Post" indicates SFT + RLVR post-training.

**Math Reasoning (MATH AVG).** We consider four diverse math benchmarks : GSM8K (Cobbe et al., 2021), MATH-500 (Hendrycks et al., 2021c), Minerva Math (Lewkowycz et al., 2022), AMC23. We report Pass@1 average of 8 runs for these.

**Science Reasoning (SCIENCE AVG).** For conceptual science and specialized knowledge, we evaluate on MMLU (Hendrycks et al., 2021a), MMLU-Pro (Wang et al., 2024), and the graduate-level STEM benchmark GPQA-Diamond (Rein et al., 2024). For science benchmarks, we report the average greedy and Pass@1 scores from 4 runs (SCIENCE AVG@1[4]).

## 4 RESULTS

Table 1 reports the performance of QWEN3-1.7B-BASE under different pretraining and post-training objectives. First, RLP consistently outperforms both the $\mathcal{M}_{\text{base}}$ and $\mathcal{M}_{\text{CPT}}$ across nearly all benchmarks, with especially strong gains on reasoning-heavy tasks such as AIME25 and MMLU-Pro. We see that $\mathcal{M}_{\text{RLP}}$ is relatively on average 19% and 17% better than $\mathcal{M}_{\text{base}}$ and $\mathcal{M}_{\text{CPT}}$ respectively. This highlights the effectiveness of dense, verifier-free reinforcement signals for instilling reasoning capabilities during pretraining. Second, the benefits of RLP persist even after strong post-training (SFT + RLVR). While all models improve after post-training, $\mathcal{M}_{\text{RLP}}$ achieves the highest scores with the overall average substantially higher than both $\mathcal{M}_{\text{base}}$ by 8% and $\mathcal{M}_{\text{CPT}}$ by 7% relatively. This indicates that RLP establishes robust reasoning foundations that are not washed out by downstream alignment but instead compound with post-training. We observe particularly large gains in science domains, with $\mathcal{M}_{\text{RLP}}$ +Post achieving +3 points over $\mathcal{M}_{\text{CPT}}$ +Post. This trend suggests that RLP is not limited to mathematical reasoning but also generalizes effectively to other domains. The ability to strengthen performance in science benchmarks highlights that RLP fosters a broader class of multi-step explanation-driven reasoning skills, moving beyond domain-specific improvements and pointing toward a more versatile foundation for reasoning in LLMs. Overall, the results demonstrate that RLP not only induces reasoning ability during pretraining but also synergizes with post-training, leading to models with stronger and more durable reasoning abilities than those trained with next-token prediction or continuous pretraining.

**Scaling Model Size and Architecture** We further scale RLP to NEMOTRON-NANO-12B-V2 (Nano, 2025) ($\mathcal{M}_{\text{base}}$), a hybrid Mamba-Transformer language model of 12B parameter size. In this comparison we take an intermediate checkpoint of NEMOTRON-NANO-12B-V2 trained till 19.8 trillion tokens and apply RLP for 250 million tokens only. $\mathcal{M}_{\text{base}}$ on the other hand is trained for 20 trillion tokens. In addition, we employ an identical post-training pipeline (SFT → RLVR), mirroring the setup used for QWEN3-1.7B-BASE in Table 1. The results as shown in Table 2 confirms that, regardless of model size and families, RLP not only yields a very large improvement at the base

stage (Overall 42.81% to 61.32%, a 43% relative gain) but that these gains persist and continue to compound after strong post training. After SFT + RLVR, the RLP trained model improves from 61.32% to 68.09%, maintaining a clear margin over the compute matched baseline (65.31%). The largest relative gains are in scientific reasoning as the Science Avg rises from 34.51% to 57.26% at the base stage and further to 64.52% after post training, compared to 58.91% for the continuously pretrained baseline. This pattern closely mirrors our findings on Qwen3, and demonstrates that RLP scales effectively both to larger parameter counts and to a different architecture family, while remaining compatible with strong downstream alignment. Furthermore, we validate that RLP scales effectively to larger backbones by applying it to QWEN3-14B-BASE, where it improves the overall average from 60.66% to 65.00% after training on 1B tokens, with particularly strong gains in scientific reasoning. Full results for this scaling experiment are provided in the Appendix 12.

**RPT Comparison**  Following the experimental setup in RPT (Dong et al., 2025), we trained $\mathcal{M}_{\text{base}}$ on both RPT and RLP methods for one epoch under tokens and flop matched compute budgets before evaluating on our benchmark suite. In the token matched setting, we trained both models on Omni-MATH (Gao et al., 2024) using the same number of input tokens. As we apply RLP to a single token per document while RPT is applied on multiple tokens per document, the number of target tokens for which reward is applied is substantially larger for RPT.

Conversely, for the flop-matched training, both models are trained on Nemotron-CrossThink (as detailed in Appendix 12) for one epoch on the same data, and we ensure that the number of target tokens for which reward is applied is same in both training runs. As summarized in Table Table 3, under *Token-Matched* setting, RLP achieves uniformly higher aggregates: *Math Avg* improves by an absolute +2.12% (+4.5% relative), *Science Avg* by +1.19% (+3.3% relative), and *Overall Avg* boosts by +1.66% (+4.0% relative). In addition, under the flop matched setting, the improvement is even more prominent. RLP achieves a 20.12% relative improvement on average over RPT. Methodologically, RPT applies reinforcement only to tokens pre-selected by an auxiliary assistant via entropy filtering and optimizes a sparse, binary next-token correctness signal that ignores the CoT content, limiting where the signal can be applied. In contrast, RLP evaluates each sampled CoT by the information gain it provides for the observed next token and updates at all positions without an auxiliary filter which yields consistently better averages under the matched setting above. Crucially, this dense, per-token information-gain reward supplies richer credit assignment than RPT's sparse binary signal and, in our matched experiments, empirically yields better performance.

| Model | Math Avg | Science Avg | Avg |
|---|---|---|---|
| *Token-Matched* | | | |
| $\mathcal{M}_{\text{RPT}}$ | 47.50 | 35.88 | 41.69 |
| $\mathcal{M}_{\textbf{RLP}}$ | **49.62** | **37.07** | **43.35** |
| *Flop-Matched* | | | |
| $\mathcal{M}_{\text{RPT}}$ | 36.66 | 34.38 | 35.68 |
| $\mathcal{M}_{\textbf{RLP}}$ | **45.95** | **38.76** | **42.86** |

Table 3: Token- and flop-matched comparisons of RLP and RPT using a QWEN3-1.7B-BASE model.

## 5 ABLATIONS

**Does RLP provide generalizable improvements across diverse corpora?**  A key advantage of RLP is its scalability to large, diverse corpora, unlike RLVR, which relies on small, curated reasoning datasets and raises concerns about generalizability. Prior work (Chen et al., 2025; Setlur et al., 2025) highlights the need for complex reasoning corpora to sustain RL improvements, but such datasets are costly to curate and impractical at pretraining scale. For these ablations, we apply RLP to QWEN3-1.7B-BASE for 200 steps—utilizing 170M input tokens—holding the rest of the setup fixed.

As illustrated in Table 4, RLP delivers consistent gains across all corpus families, eliminating concerns that RL based pretraining only benefits curated reasoning data. Relative to $\mathcal{M}_{\text{base}}$ average improves by 7-9% with strongest gains on Nemotron-Crossthink (SFT-style) and Web-Crawl (general-purpose corpora). Unlike prior work (Akter et al., 2025), where RL gains were limited to math and weakened under mixed data, RLP achieves simultaneous improvements across all benchmarks, demonstrating genuine cross-domain transfer. Even on purely non-reasoning general corpora such as web-crawl, RLP extracts a reasoning signal that scales with data diversity (Appendix 12). Table 4 illustrates that unlike prior work (Liu et al., 2025b; Zhou et al., 2025), RLP can be applied to any

| Model | Dataset | Type | Math Avg | Science Avg | Science Avg@1[4] | Avg |
|---|---|---|---|---|---|---|
| $\mathcal{M}_{\text{base}}$ | - | - | 35.96 | 34.50 | 32.11 | 34.19 |
| $\mathcal{M}_{\text{CPT}}$ | Nemotron-Crossthink [170M] | Equal Input Token | 37.11 | 35.76 | 32.15 | 35.01 |
| | Nemotron-Crossthink [6B] | Equal FLOPs | 43.90 | 37.74 | 32.47 | 38.04 |
| | $\mathcal{D}_{\text{PT}}$[1B] | PT Data Mix | 45.34 | 32.14 | 29.33 | 35.60 |
| $\mathcal{M}_{\text{RLP}}$ | OmniMath [170M] | | 46.48 | 40.27 | 37.54 | 41.43 |
| | OpenThoughts [170M] | SFT | 47.64 | 40.84 | 35.88 | 41.45 |
| | Nemotron-Crossthink [170M] | | 49.76 | 42.54 | 37.78 | **43.36** |
| | ACAD [170M] | | 47.68 | 40.59 | 36.87 | 41.71 |
| | Math-Text [170M] | General | 48.07 | 40.46 | 36.32 | 41.62 |
| | Web-Crawl [170M] | | 48.87 | 40.75 | 36.77 | **42.13** |
| | $\mathcal{D}_{\text{PT}}$[1B] | PT Data Mix | 46.35 | 39.68 | 36.67 | 40.90 |

Table 4: **RLP across diverse corpora.** RLP trained on six SFT-style and general-purpose datasets yields consistent gains, indicating transferable reasoning from mixed/open-ended data.

data format like academic papers, textbooks, web-crawl as well as SFT style data. Overall, RLP is scalable, domain-agnostic pre-training augmentation that enhances both reasoning and accuracy.

**Does the improvement sustain under compute equivalent baselines?** A critical question is whether RLP's gains stem from its unique RL-based pretraining or simply higher compute. Standard next-token pretraining quantifies compute by input tokens, but RLP adds rollout costs not captured by this metric. For fair comparison, we evaluate against $\mathcal{M}_{\text{CPT}}$ baselines under: (a) equal Input Tokens Seen and (b) equal total Compute FLOPs. RLP is fixed to $T_{inp} = 170M$ tokens; the token-matched $\mathcal{M}_{\text{CPT}}$ [170M] continues pretraining on 170M tokens (Input Token), while the FLOP-matched budget corresponds to 6B tokens for CPT ($\mathcal{M}_{\text{CPT}}$ [6B])(see Appendix 11).

In Table 4, $\mathcal{M}_{\text{RLP}}$ outperforms $\mathcal{M}_{\text{CPT}}$ trained on the same 170M tokens and maintains a clear advantage even against a compute-matched $\mathcal{M}_{\text{CPT}}$ exposed to 6B tokens (35× more data). Despite this disparity, RLP achieves a 5.3% gain on average (compare $\mathcal{M}_{\text{CPT}}$ Nemotron-Crossthink [6B] vs $\mathcal{M}_{\text{RLP}}$ Nemotron-Crossthink [170M]), with consistent improvements across math and science benchmarks. These results show that RLP's gains stem not from more efficient use of compute, not larger budgets, validating the effectiveness of our approach.

**Is RLP comparable to CPT with high-quality reasoning data?** High-quality reasoning corpora have shown to substantially boost base model reasoning ability when used in continuous pretraining (CPT) or mid-training (Wang et al., 2025; Gandhi et al., 2025). This raises the important question of whether CPT can match or even surpass RLP under such favorable conditions. To investigate this, we conduct CPT on both reasoning-centric, Nemotron-Crossthink and general pretraining ($\mathcal{D}_{\text{PT}}$) datasets, each using 170M tokens. Our results in Table 4 show that even with high quality reasoning data, RLP consistently outperforms CPT by a significant margin. Specifically, $\mathcal{M}_{\text{RLP}}$ outperforms $\mathcal{M}_{\text{CPT}}$, showing an average gain of 8% on Nemotron-Crossthink and 5% on pre-training data mix ($\mathcal{D}_{\text{PT}}$) on 1B tokens. These results highlight two key insights. First, while CPT benefits from reasoning-dense corpora, it remains sensitive to domain skew—evident in the weak science accuracy on $\mathcal{D}_{\text{PT}}$—whereas RLP generalizes more evenly across disciplines. Second, the consistent margin by which RLP outperforms CPT, even in the presence of high quality reasoning data, underscores that the gains of RLP are not merely due to data quality but stem from the algorithmic design itself. This reinforces the conclusion that RLP provides a generalizable mechanism for leveraging reasoning data during pretraining, complementing rather than being overshadowed by high-quality corpus selection.

**Ablations on rollout count, completion length, and KL weight.** Fig. 2 visualizes the trends across three settings: (a) rollouts, (b) completion length, and (c) KL. Please look into §11 for more detailed numbers and per-task breakdowns. More rollouts help up to $G = 16$ (*Overall* 42.17%); $G = 4$ and 8 already reach 41.38% and 41.95%, while $G = 32$ decreases slightly to 41.75% (Fig. 2a). Increasing completion length gives the largest gains. Specifically, *overall* rises from 11.50% at 64 to 42.17% at 2048, with *Math/Science* moving from 1.12%/21.88% to 48.06%/36.29% (Fig. 2b). Extending to 4096 yields 42.21% at roughly twice the thought budget, so we default to 2048. Furthermore, a KL anchor does not help. Specifically, $\beta = 10^{-4}$ and $10^{-3}$ give 41.35% and 41.44%, compared to

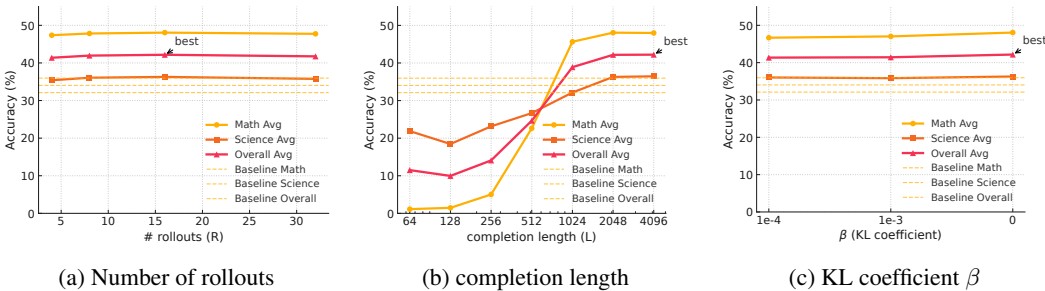

| (a) Number of rollouts | (b) completion length | (c) KL coefficient $\beta$ |

Figure 2: **Ablations on Qwen3-1.7B.** Curves report Math/Science/Overall averages. Dashed lines mark the base model.

42.17% at $\beta = 0$, and it also increases memory and step time (Fig. 2c). We therefore use $G = 16$, completion length 2048, and $\beta = 0$ in later experiments.

## 6 RELATED WORK

**Next-Token Prediction.** Next-token prediction is the standard pretraining objective for LLMs: predict the next word from prior context (Shannon, 1951; Bengio et al., 2003). Scaling it with Transformers (Vaswani et al., 2017) enabled landmark and state-of-the-art systems (Radford et al., 2018; Brown et al., 2020; Smith et al., 2022; Bi et al., 2024; Nano, 2025; Yang et al., 2025). Anticipating tokens across corpora induces syntactic, semantic, and pragmatic structure that transfers broadly. Alternatives include masked language modeling (Devlin et al., 2019) and span corruption (Raffel et al., 2020), but next-token prediction remains dominant for its alignment with left-to-right generation and strong downstream accuracy across tasks. In this work. we add a verifier-free dense reward during pretraining that leverages reasoning before prediction.

**Verifier-Free Rewards in Post-Training.** Recent work explores verifier-free rewards. Yuan et al. (2024) uses iterative DPO where, after SFT, the model judges its own candidates to create preference pairs. Liu et al. (2025b) trains with incentive RL on SFT corpora. Zhao et al. (2025) proposes RL from an internal feedback while using the model's confidence as reward. RLP, in contrast, is a GRPO-style pretraining objective. It operates on any text data including web-crawl, academic papers and SFT datasets and optimizes continuation quality beyond next-token prediction. Because these methods target post-training policies, direct comparisons are not well-posed.

## 7 CONCLUSION

We introduce RLP, a reinforcement pretraining objective that rewards chain-of-thought by its information gain for next-token prediction. Unlike traditional approaches that defer RL to post-training, RLP instills reasoning during pretraining, yielding gains that persist and compound after alignment. Experiments across datasets, domains, and architectures show that RLP consistently outperforms compute-matched baselines and scales efficiently to large hybrid models, establishing reinforcement pretraining as a principled and general alternative to likelihood-only training.

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

## 8 APPENDIX

### 8.1 PROOFS

In this section, we provide the proofs supporting the methodology in §2. We first prove the tokenwise cross-entropy (CE) reduction identity (Prop. 1), then the lower bound via marginalization over thoughts (Prop. 2). Finally, we state and prove Prop. 3, which formalizes the positionwise-credit claim described in §2.5: under teacher forcing, averaging the expected tokenwise information-gain rewards across positions recovers the expected per-token sequence-level CE improvement.

For convenience, we recall the key definitions from the main text: the reasoned and baseline log-evidence $S_{\text{pred}}(c_t) = \log p_\theta(x_t \mid x_{<t}, c_t)$ and $S_{\text{EMA}} = \log \bar{p}_\phi(x_t \mid x_{<t})$ (equation 3); the information-gain reward $r(c_t) = S_{\text{pred}}(c_t) - S_{\text{EMA}}$ (equation 5); and the cross-entropy $\text{CE}(p, q) \overset{\text{def}}{=} \mathbb{E}_{x \sim p}\big[-\log q(x)\big]$ (equation 2).

### 8.2 PROOF OF PROPOSITION 1 (EXPECTED IMPROVEMENT IDENTITY)

*Proof of Proposition 1.* Fix the context $x_{<t}$ and a realized thought $c_t$, and let $p_t^*(x) := p^*(x \mid x_{<t})$ denote the data distribution over $x_t$ at this position. By the reward definition equation 5 together with equation 3,

$$r(c_t) = \log p_\theta\big(x_t \mid x_{<t}, c_t\big) - \log \bar{p}_\phi\big(x_t \mid x_{<t}\big).$$

Taking expectation with respect to $x_t \sim p_t^*$ and using linearity of expectation,

$$\underset{x_t \sim p_t^*}{\mathbb{E}}\big[r(c_t)\big] = \underset{x_t \sim p_t^*}{\mathbb{E}}\Big[\log p_\theta(x_t \mid x_{<t}, c_t)\Big] - \underset{x_t \sim p_t^*}{\mathbb{E}}\Big[\log \bar{p}_\phi(x_t \mid x_{<t})\Big].$$

By the definition of cross-entropy equation 2, $\text{CE}(p, q) = \mathbb{E}_{x \sim p}[-\log q(x)]$, so each expectation of a log-likelihood equals the negative cross-entropy:

$$\underset{x_t \sim p_t^*}{\mathbb{E}}\Big[\log p_\theta(x_t \mid x_{<t}, c_t)\Big] = -\text{CE}\big(p^*, p_\theta(\cdot \mid x_{<t}, c_t)\big), \qquad \underset{x_t \sim p_t^*}{\mathbb{E}}\Big[\log \bar{p}_\phi(x_t \mid x_{<t})\Big] = -\text{CE}\big(p^*, \bar{p}_\phi(\cdot \mid x_{<t})\big).$$

Substituting into the previous display yields

$$\underset{x_t \sim p_t^*}{\mathbb{E}}\big[r(c_t)\big] = \text{CE}\big(p^*, \bar{p}_\phi(\cdot \mid x_{<t})\big) - \text{CE}\big(p^*, p_\theta(\cdot \mid x_{<t}, c_t)\big),$$

which is the desired identity. □

### 8.3 PROOF OF PROPOSITION 2 (LOWER BOUND VIA MARGINALIZATION OVER THOUGHTS)

*Proof of Proposition 2.* Fix $(x_{<t}, x_t)$ and recall $S_{\text{pred}}(c_t) = \log p_\theta(x_t \mid x_{<t}, c_t)$ and $\tilde{p}_\theta(x \mid x_{<t}) = \mathbb{E}_{z_t \sim \pi_\theta(\cdot \mid x_{<t})}\big[p_\theta(x \mid x_{<t}, z_t)\big]$.

**(i) Jensen bound.** Conditioning on $(x_{<t}, x_t)$ and taking expectation over $c_t \sim \pi_\theta(\cdot \mid x_{<t})$,

$$\mathbb{E}_{c_t \sim \pi_\theta}\big[S_{\text{pred}}(c_t)\big] = \mathbb{E}_{c_t}\Big[\log p_\theta(x_t \mid x_{<t}, c_t)\Big] \leq \log \mathbb{E}_{c_t}\Big[p_\theta(x_t \mid x_{<t}, c_t)\Big] = \log \tilde{p}_\theta(x_t \mid x_{<t}),$$

where the inequality is Jensen's inequality applied to the concave function $\log(\cdot)$. This proves (i) pointwise for the realized $x_t$.

**(ii) Bound on $J(\theta)$.** By definition of the reward in equation 5 and teacher forcing (see equation 3),

$$J(\theta) = \mathbb{E}\Big[\mathbb{E}_{c_t \sim \pi_\theta}\big[S_{\text{pred}}(c_t)\big] - S_{\text{EMA}}\Big] \leq \mathbb{E}\Big[\log \tilde{p}_\theta(x_t \mid x_{<t}) - \log \bar{p}_\phi(x_t \mid x_{<t})\Big] = \mathbb{E}\left[\log \frac{\tilde{p}_\theta(x_t \mid x_{<t})}{\bar{p}_\phi(x_t \mid x_{<t})}\right],$$

where the inequality uses part (i) and the outer expectation is over $(x_{<t}, x_t) \sim \mathcal{D}$. This proves (ii).

**Tightness.** Equality in (i) (and hence in (ii)) holds precisely when $p_\theta(x_t \mid x_{<t}, c_t)$ is almost surely constant in $c_t$ under $\pi_\theta(\cdot \mid x_{<t})$ (e.g., when the predictor ignores the thought or when the thought policy is degenerate).

□

### 8.4 TOKENWISE–TO–SEQUENCE CONNECTION UNDER TEACHER FORCING (POSITIONWISE CREDIT)

This subsection formalizes the claim in §2.5 that summing positionwise CE improvements recovers the sequence-level (per-token) improvement. The following proposition is *new to the appendix* and not required elsewhere; it clarifies how tokenwise rewards aggregate at the sequence level under teacher forcing.

**Proposition 3** (Tokenwise–to–sequence connection under teacher forcing). *Let $\boldsymbol{x} = (x_1, \ldots, x_T)$ be drawn from the data distribution $p^*(\boldsymbol{x})$ and fix a policy $\pi_\theta(c_t \mid x_{<t})$, the reasoned scorer $p_\theta(\cdot \mid x_{<t}, c_t)$, and the no-think baseline $\bar{p}_\phi(\cdot \mid x_{<t})$. Define the sequence-level (per-token) cross-entropy for the baseline and the (stochastic) reasoned scorer by*

$$\mathrm{CE}_{\mathrm{seq}}(p^*, \bar{p}_\phi) := \mathbb{E}_{\boldsymbol{x} \sim \mathcal{D}} \left[ -\frac{1}{T} \sum_{t=1}^T \log \bar{p}_\phi(x_t \mid x_{<t}) \right],$$

$$\mathrm{CE}_{\mathrm{seq}}(p^*, p_\theta[\pi_\theta]) := \mathbb{E}_{\boldsymbol{x} \sim \mathcal{D}} \left[ -\frac{1}{T} \sum_{t=1}^T \mathbb{E}_{c_t \sim \pi_\theta(\cdot \mid x_{<t})} \left[ \log p_\theta(x_t \mid x_{<t}, c_t) \right] \right].$$

*Then the average over positions of the* expected *tokenwise information-gain rewards equals the per-token sequence-level CE improvement of the reasoned scorer against the baseline:*

$$\mathbb{E}_{\boldsymbol{x}} \left[ \frac{1}{T} \sum_{t=1}^T \mathbb{E}_{c_t \sim \pi_\theta(\cdot \mid x_{<t})} \mathbb{E}_{x_t \sim p^*(\cdot \mid x_{<t})} [r(c_t)] \right] = \mathrm{CE}_{\mathrm{seq}}(p^*, \bar{p}_\phi) - \mathrm{CE}_{\mathrm{seq}}(p^*, p_\theta[\pi_\theta]).$$

*Proof.* **(i) Conditional independence under teacher forcing.** At position $t$, teacher forcing samples the target token from the data channel while the thought is sampled from the policy given the same prefix:

$$x_t \sim p^*(\cdot \mid x_{<t}), \qquad c_t \sim \pi_\theta(\cdot \mid x_{<t}).$$

Hence

$$p(c_t, x_t \mid x_{<t}) = \pi_\theta(c_t \mid x_{<t}) \, p^*(x_t \mid x_{<t}), \quad \text{i.e.} \quad c_t \perp x_t \mid x_{<t}.$$

This implies $\mathbb{E}_{x_t \sim p^*(\cdot \mid x_{<t}, c_t)}[\cdot] = \mathbb{E}_{x_t \sim p^*(\cdot \mid x_{<t})}[\cdot]$.

**(ii) Positionwise CE reduction.** By Proposition 1, for any fixed $(x_{<t}, c_t)$,

$$\mathbb{E}_{x_t \sim p^*(\cdot \mid x_{<t})} [r(c_t)] = \mathrm{CE}(p^*, \bar{p}_\phi(\cdot \mid x_{<t})) - \mathrm{CE}(p^*, p_\theta(\cdot \mid x_{<t}, c_t)).$$

Taking expectation over $c_t \sim \pi_\theta(\cdot \mid x_{<t})$ and using linearity of expectation gives

$$\mathbb{E}_{c_t} \mathbb{E}_{x_t} [r(c_t)] = \mathrm{CE}(p^*, \bar{p}_\phi(\cdot \mid x_{<t})) - \mathbb{E}_{c_t} \mathrm{CE}(p^*, p_\theta(\cdot \mid x_{<t}, c_t)).$$

**(iii) Sum over positions.** Average the identity in (ii) over $t = 1, \ldots, T$ and over $\boldsymbol{x} \sim \mathcal{D}$:

$$\mathbb{E}_{\boldsymbol{x}} \left[ \frac{1}{T} \sum_{t=1}^T \mathbb{E}_{c_t} \mathbb{E}_{x_t} [r(c_t)] \right]$$

$$= \mathbb{E}_{\boldsymbol{x}} \left[ \frac{1}{T} \sum_{t=1}^T \mathrm{CE}(p^*, \bar{p}_\phi(\cdot \mid x_{<t})) \right] - \mathbb{E}_{\boldsymbol{x}} \left[ \frac{1}{T} \sum_{t=1}^T \mathbb{E}_{c_t} \mathrm{CE}(p^*, p_\theta(\cdot \mid x_{<t}, c_t)) \right].$$

By the definition of cross-entropy in equation 2 and the chain rule for likelihoods,

$$\mathbb{E}_{x_t \sim p^*(\cdot \mid x_{<t})} [-\log \bar{p}_\phi(x_t \mid x_{<t})] = \mathrm{CE}(p^*, \bar{p}_\phi(\cdot \mid x_{<t})),$$

and similarly for the reasoned scorer inside the $c_t$-expectation. Therefore the two sums on the right are exactly $\mathrm{CE}_{\mathrm{seq}}(p^*, \bar{p}_\phi)$ and $\mathrm{CE}_{\mathrm{seq}}(p^*, p_\theta[\pi_\theta])$ as defined above, yielding the claimed equality.

$\square$

# 9 WHY RELATIVE ADVANTAGES DO NOT REWARD BAD THOUGHTS

## 9.1 PROOF OF MONOTONIC IMPROVEMENT

It may seem paradoxical that, when all thoughts perform poorly ($r(c_t) < 0$), the group-relative formulation still labels one as "better" and reinforces it. Does this mean the model is being trained to favor bad reasoning? We demonstrate that, mathematically, this mechanism is sound: the update remains an unbiased gradient step on $J(\theta)$, ensuring monotonic improvement even in such cases.

**1.Objective.** For context $x_{<t}$ and target token $x_t$:

$$J(\theta) = \mathbb{E}_{c \sim \pi_\theta}\big[r(c)\big], \qquad r(c) = \log p_\theta(x_t \mid x_{<t}, c) - \log \bar{p}_\phi(x_t \mid x_{<t}). \tag{9}$$

Maximizing $J$ reduces cross-entropy versus the no-think baseline. Ignoring stop-gradients, the policy gradient is

$$\nabla_\theta J(\theta) = \mathbb{E}_{c \sim \pi_\theta}\big[r(c)\,\nabla_\theta \log \pi_\theta(c)\big]. \tag{10}$$

**2. Group-relative advantages are unbiased.** We draw $G \geq 2$ thoughts $c^{(1)}, \ldots, c^{(G)} \sim \pi_\theta$ and form

$$\bar{r} = \tfrac{1}{G}\sum_{j=1}^{G} r(c^{(j)}), \tag{11}$$

$$A^{(i)} = \tfrac{G}{G-1}\Big(r(c^{(i)}) - \bar{r}\Big). \tag{12}$$

Let $\mu = \mathbb{E}[r(c)]$. Then

$$\mathbb{E}[A^{(i)} \mid c^{(i)}] = r(c^{(i)}) - \mu, \tag{13}$$

and

$$\mathbb{E}\left[\frac{1}{G}\sum_{i=1}^{G} A^{(i)}\nabla_\theta \log \pi_\theta(c^{(i)})\right] = \nabla_\theta J(\theta). \tag{14}$$

Hence, the estimator is **unbiased**. Even if all rewards are negative, the update follows the correct gradient direction.

**3. Why positive advantage for the "least-bad" rollout is correct.** As the model learns, it gradually increases the probability of generating thoughts that help prediction and decreases the probability of those that do not. This process, known as the *replicator dynamic*, captures how relative advantages drive steady improvement over time:

$$\dot{\pi}(c) = \pi(c)\,[r(c) - \mu], \tag{15}$$

whose improvement rate is

$$\frac{d}{d\tau}J(\theta(\tau)) = \mathrm{Var}_\pi[r(c)] \geq 0. \tag{16}$$

Even if all $r(c) < 0$, shifting probability mass from more-negative to less-negative thoughts *increases* $J$. Thus, a positive advantage for the least-bad thought reflects correct relative improvement, not misaligned reward.

**4. Monotonic expected improvement.** With unbiased gradient estimator $\hat{g} \approx \nabla J$ and small step size $\alpha$:

$$\mathbb{E}[J(\theta + \alpha\hat{g})] \approx J(\theta) + \alpha\|\nabla J(\theta)\|_2^2 \geq J(\theta), \tag{17}$$

ensuring monotonic improvement in expectation.

**5. The gradient does not blindly increase harmful thoughts.** A remaining concern is that a thought with negative reward $r(c) < 0$ might still receive a positive advantage $A(c) > 0$ if it is simply less harmful than its peers, apparently encouraging bad reasoning. However, the gradient update does not blindly amplify such thoughts; it reallocates probability mass among them in a way that *improves the expected objective*.

First, because the advantages are defined as

$$A(c) = \tfrac{G}{G-1}\big(r(c) - \bar{r}\big), \qquad \text{with} \quad \bar{r} = \tfrac{1}{G}\sum_{j=1}^{G} r(c^{(j)}),$$

the total $\sum_i A(c^{(i)}) = 0$. Hence, even if every reward is negative, the update is zero-sum: probability increases only for thoughts that are *less negative* than average, while it decreases for those that are worse. This shift raises the expected reward $J(\theta)$ because the expected improvement rate is

$$\frac{d}{d\tau} J(\theta(\tau)) = \mathrm{Var}_\pi[r(c)] \geq 0.$$

Thus, the method performs a relative reallocation and guarantees monotonic ascent in expectation.

Second, a positive advantage $A(c) > 0$ does not deterministically increase the corresponding $r(c)$ on the next update; it increases it *in expectation*. The policy gradient on thought tokens,

$$\nabla_\theta \mathcal{L}_{\mathrm{IG}} \propto -A(c)\,\nabla_\theta \log \pi_\theta(c),$$

acts on the relative usefulness of each thought, not its absolute reward value. Over repeated steps, the model raises the log-evidence $\log p_\theta(x_t \mid x_{<t}, c)$ for those thoughts that contribute more to prediction, thereby increasing their expected $r(c)$ relative to the slowly moving EMA baseline $\bar{p}_\phi$.

Third, the EMA baseline prevents artificial reward inflation. Because $\bar{p}_\phi$ lags behind $\theta$ through a slow exponential moving average, any transient or spurious improvement in $r(c)$ dissipates as the baseline catches up. Sustained positive advantages arise only when the model genuinely improves predictive likelihood relative to the no-think counterfactual.

Finally, while a positive advantage can momentarily reinforce a thought whose raw reward remains negative, this update is not pathological. It simply redirects probability toward the least harmful reasoning pattern available, reducing overall loss. Over time, these relatively better thoughts typically evolve into genuinely helpful ones as their predictive evidence increases, ensuring that the training process remains stable and aligned with maximizing $J(\theta)$.

## 9.2 NUMERICAL ILLUSTRATION OF RELATIVE ADVANTAGE UPDATES

To make the abstract dynamics more concrete, we present a simple numerical example showing how the group-relative advantage mechanism improves the expected objective $J(\pi; r)$ even when all rewards are initially negative. Note that in this illustrative example we denote the expected reward as $J(\pi; r)$ to emphasize its dependence on the discrete policy over thoughts $\pi$ and fixed rewards $r_i$. Conceptually, this corresponds to the same information-gain objective $J(\theta)$ introduced in the main text, expressed here in a simplified form.

We consider four sampled thoughts $c_1, c_2, c_3, c_4$ with policy $\pi = [\pi_1, \pi_2, \pi_3, \pi_4]$, initialized uniformly. For each thought, the information-gain reward is

$$r_i = \log p_\theta(x_t \mid x_{<t}, c_i) - \log \bar{p}_\phi(x_t \mid x_{<t}),$$

and the group size is $G = 4$ with mean reward $\bar{r} = \tfrac{1}{4}\sum_i r_i$. The group-relative advantage is

$$A_i = \tfrac{G}{G-1}(r_i - \bar{r}) = \tfrac{4}{3}(r_i - \bar{r}),$$

and we assume each thought has length $|c_i| = 4$ so that per-token weight is $A_i/4$. The policy is updated by an exponentiated-gradient (replicator) step

$$\pi_{\mathrm{new}}(i) \propto \pi_{\mathrm{old}}(i)\,\exp(\eta A_i), \quad \text{with } \eta = 0.5,$$

and the expected objective is $J(\pi; r) = \sum_i \pi_i r_i$.

Although a positive advantage can momentarily reinforce a thought whose raw reward $r_i$ is still negative, this update is not pathological. Because advantages are computed relative to the group mean, a positive $A_i$ simply indicates that $c_i$ is *less harmful* than its peers. Increasing its probability reallocates mass away from worse alternatives, thereby improving the expected objective $J$. Over subsequent updates, the model typically adapts to make these less-harmful thoughts genuinely helpful, raising $r_i$ in expectation.

**Iteration 1: all thoughts are harmful ($r_i < 0$), but one is least bad.**

$$\pi^{(0)} = [0.25, 0.25, 0.25, 0.25], \qquad r^{(1)} = [-0.80, -0.60, -0.50, -0.30].$$

Mean and advantages:

$$\bar{r}^{(1)} = -0.55, \qquad A^{(1)} = [-0.3333, -0.0667, +0.0667, +0.3333].$$

Per-token weights: $A^{(1)}/|c| = [-0.0833, -0.0167, +0.0167, +0.0833]$. Note that $c_4$ has $r_4 = -0.30 < 0$ yet receives a positive advantage $A_4 = +0.3333$, so every token in $c_4$ gets a positive gradient. Policy update with $\eta = 0.5$ gives

$$\pi^{(1)} \propto \pi^{(0)} \odot \exp(0.5 A^{(1)}) = [0.2101, 0.2401, 0.2566, 0.2932],$$

yielding $J(\pi^{(0)}; r^{(1)}) = -0.5500$ and $J(\pi^{(1)}; r^{(1)}) = -0.5284$. This is a small but consistent improvement.

**Iteration 2: dense updates improve $r$ on $c_3, c_4$.**

$r^{(2)} = [-0.80, -0.60, -0.35, -0.10], \qquad \bar{r}^{(2)} = -0.4625, \qquad A^{(2)} = [-0.4500, -0.1833, +0.1500, +0.4833].$

Update:

$$\pi^{(2)} \propto \pi^{(1)} \odot \exp(0.5 A^{(2)}) = [0.1618, 0.2113, 0.2668, 0.3601].$$

Expected objective: $J(\pi^{(1)}; r^{(2)}) = -0.4313, J(\pi^{(2)}; r^{(2)}) = -0.3856$.

**Iteration 3: the least-bad thought becomes genuinely helpful.**

$r^{(3)} = [-0.80, -0.60, -0.20, +0.05], \qquad \bar{r}^{(3)} = -0.3875, \qquad A^{(3)} = [-0.5500, -0.2833, +0.2500, +0.5833].$

Policy update:

$$\pi^{(3)} \propto \pi^{(2)} \odot \exp(0.5 A^{(3)}) = [0.1127, 0.1681, 0.2772, 0.4420],$$

and the expected objective improves again: $J(\pi^{(2)}; r^{(3)}) = -0.2916, J(\pi^{(3)}; r^{(3)}) = -0.2244$.

As seen in the above, in Iteration 1, all rewards are negative, yet $c_4$ (the least bad)has a positive advantage, showing how the dense loss pushes probability toward less harmful thoughts and increases $J$. Since rewards are tied to log-evidence, these positive gradients directly improve the corresponding $r(c)$ values, leading to less-negative and eventually positive rewards in later iterations.

## 10 EXPERIMENTAL SETUP

**RLP:** We employ RLP on both base and intermediate checkpoints using diverse datasets. To facilitate this, we use Hugging Face (2025) as the RL training backbone and deploy training using 32 H100 80GB SXM5 GPUs for 170M to 10B tokens. We train the base models with key settings including a constant learning rate of $1e^{-6}$, a batch size of 512 and a maximum context length of 2048 tokens. Each generation step contains 512 unique prompts sampled from the dataset, and performing 16 rollouts with temperature 0.7. We set KL coefficient to 0 across all runs.

**Continuous Pre-training:** We continuously pretrain the $\mathcal{M}_{\text{base}}$ model using both general pre-training and specialized post-training corpus to draw comparison between pretraining and RLP training objective. For this experimentation, we use Megatron-LM (Shoeybi et al., 2019) as the pretraining backbone and continuously train on 32 H100 80GB SXM5 GPUs for 170M to 10B tokens depending on the data size and comparison requirement. During training, we use the AdamW optimizer (Loshchilov & Hutter, 2019) with $\beta_1 = 0.9$, $\beta_2 = 0.95$ and weight decay of 0.1. We use a 2-way tensor and pipeline parallelism to train the model. We set the maximum value of learning rate to $1e^{-6}$, minimum to $1e^{-7}$, and use a batch size of 6M tokens with a 8192 context length.

**Post-Training:** For supervised fine-tuning (SFT), we use the OpenThoughts3 dataset (Guha et al., 2025). We filtered examples that did not include a final answer. With this filtering scheme, the total number of samples for SFT post-training is $45, 6024$. For RLVR, we used the The Mathematics Aptitude Test of Heuristics (MATH) dataset (Hendrycks et al., 2021b) with $7, 500$ examples. This dataset includes problems from various subjects such as algebra, geometry, number theory and precalculus. We trained models in all RLVR experiments for 1 epoch with a global batch size of 1024 and used cosine annealing and an initial learning rate of $1e^{-6}$.

**Prompt** Given a context $x_{<t}$, we ask the model to reason about the target token $x_t$ using the following prompt, $p$.

> **System Prompt, $p$**
>
> "You are a continuation-and-reasoning assistant. You receive the prefix of a context, problem, solution, or derivation. First, briefly think between $</\text{think}>$ and $</\text{think}>$ about what should come next. Then, after $</\text{think}>$, continue the text in the SAME style as the prefix (notation, LaTeX, tone), focusing on the next few steps rather than jumping to a final boxed answer. Do not restate the question or add meta commentary; simply continue the content."

## 11 EXTENDED ABLATION DETAILS

Table S.1 reports per-task accuracies for each setting, and Fig. 2 provides the corresponding curves for (a) rollout count, (b) completion length, and (c) KL coefficient. Unless stated, each sweep holds the other two dimensions at the best configuration (16 rollouts, completion length 2048, $\beta = 0$).

**Rollout count.** Increasing $G$ improves accuracy up to $G = 16$, where *Overall* reaches $42.17\%$ (from $34.03\%$, $+8.14$ points). The largest taskwise lifts at $G = 16$ relative to the base are GSM8K ($+22.96$), MATH-500 ($+13.85$), MIVA ($+7.20$), MMLU ($+6.35$), and MMLU-PRO ($+6.20$), while GPQA is unchanged ($27.51$ vs $27.52$). Moving from $G = 16$ to $G = 32$ slightly lowers *Overall* to $41.75$ ($-0.42$), driven mainly by GPQA ($-2.13$), with other tasks nearly flat (e.g., MMLU-PRO $+0.79$, MMLU $-0.24$). This suggests diminishing returns once the group-relative estimator is already well-sampled.

**Completion length.** Capacity on the thought channel dominates performance. Very short completions underperform sharply: at length 64, *Overall* is $11.50$ and *Math* averages $1.12$. Increasing to 512 raises *Overall* to $24.65$ and *Math* to $22.63$. The main jump occurs between 512 and 1024 (*Overall* $+14.24$ to $38.89$; GSM8K $+28.55$; MATH-500 $+36.85$). Extending to 2048 adds a smaller but consistent gain (*Overall* $42.17$, $+3.28$ over 1024; *Math/Science* $48.06/36.29$). Pushing to 4096 gives only a marginal change (*Overall* $42.21$, $+0.04$; small taskwise shifts such as MMLU-PRO $+0.64$ and GSM8K $-0.62$), so 2048 is the preferred trade-off.

**KL coefficient.** Adding a token-level KL toward a fixed reference does not help overall. At $\beta = 10^{-4}$ and $10^{-3}$, *Overall* is $41.35$ and $41.44$ ($-0.82$ and $-0.73$ vs $\beta = 0$). There are isolated improvements (MMLU-PRO $+1.43$ at $10^{-4}$; AMC23 $+1.88$ at $10^{-3}$), but these are offset by broader declines (e.g., GSM8K $-1.26$ and $-2.82$; GPQA $-2.01$ and $-1.51$). The KL term also increases memory use and step time. We therefore keep $\beta = 0$ in the main recipe.

In summary, the appendix table provides the taskwise breakdown behind these trends, and the figure shows the smooth saturation with rollouts, the strong length-driven regime change between 512 and 1024 tokens, and the lack of net benefit from KL.

## 12 ADDITIONAL ABLATIONS

**Does the improvement sustain if we make Pretraining compute equivalent to RLP?** For both comparisons, the configuration for RLP remains fixed, based on a budget of $T_{inp} = 170M$ input tokens. First, we establish a baseline by continuing the pretraining of the base model on an identical 170M tokens (Base + CPT, Input Token). Second, to create a FLOP-equivalent baseline, we first approximate the total computational cost of RLP. The effective token budget, $T_{flop}$, can be estimated

Table S.1: Ablations on rollout count, completion length, and KL weight $\beta$ with QWEN3-1.7B-BASE. All numbers denote accuracy (%).

| Model / Variant | Tasks (%) | | | | | | | Macro avg (%) | | |
|---|---|---|---|---|---|---|---|---|---|---|
| | MATH500 | GSM8K | AMC23 | Minerva | MMLU | MMLU-Pro | GPQA | Math | Science | Overall |
| *Baseline* | | | | | | | | | | |
| Qwen3-1.7B-Base | 48.45 | 54.16 | 25.94 | 15.30 | 44.85 | 23.95 | 27.52 | 35.96 | 32.11 | 34.03 |
| *Ablation: # rollouts* | | | | | | | | | | |
| num_rollouts=4 | 59.45 | 74.79 | 33.44 | 21.78 | 50.83 | 28.81 | 26.52 | 47.37 | 35.39 | 41.38 |
| num_rollouts=8 | 61.70 | 76.93 | 30.62 | 22.06 | 50.88 | 30.55 | 26.77 | 47.83 | 36.07 | 41.95 |
| num_rollouts=16$^{\dagger}$ | 62.30 | 77.12 | 30.31 | 22.50 | 51.20 | 30.15 | 27.51 | 48.06 | 36.29 | **42.17** |
| num_rollouts=32 | 60.45 | 77.26 | 30.94 | 22.29 | 50.96 | 30.94 | 25.38 | 47.74 | 35.76 | 41.75 |
| *Ablation: completion length* | | | | | | | | | | |
| completion_length=64 | 1.00 | 2.84 | 0.62 | 0.00 | 33.26 | 15.46 | 16.92 | 1.12 | 21.88 | 11.50 |
| completion_length=128 | 1.73 | 3.17 | 0.94 | 0.05 | 29.04 | 13.94 | 12.37 | 1.47 | 18.45 | 9.96 |
| completion_length=256 | 2.95 | 13.86 | 2.81 | 0.46 | 37.19 | 17.09 | 15.15 | 5.02 | 23.14 | 14.08 |
| completion_length=512 | 21.35 | 46.58 | 16.25 | 6.34 | 42.27 | 19.82 | 17.93 | 22.63 | 26.67 | 24.65 |
| completion_length=1024 | 58.20 | 75.13 | 28.80 | 20.47 | 48.36 | 27.74 | 20.31 | 45.65 | 32.14 | 38.89 |
| completion_length=2048$^{\dagger}$ | 62.30 | 77.12 | 30.31 | 22.50 | 51.20 | 30.15 | 27.51 | 48.06 | 36.29 | 42.17 |
| completion_length=4096 | 62.00 | 76.50 | 30.60 | 22.80 | 51.30 | 30.79 | 27.27 | 47.98 | 36.45 | **42.21** |
| *Ablation: KL weight $\beta$* | | | | | | | | | | |
| $\beta = 10^{-4}$ | 61.35 | 75.86 | 28.00 | 21.50 | 51.00 | 31.58 | 25.50 | 46.68 | 36.03 | 41.35 |
| $\beta = 10^{-3}$ | 60.90 | 74.30 | 32.19 | 20.73 | 50.73 | 30.80 | 26.00 | 47.03 | 35.84 | 41.44 |
| $\beta = 0^{\dagger}$ | 62.30 | 77.12 | 30.31 | 22.50 | 51.20 | 30.15 | 27.51 | 48.06 | 36.29 | **42.17** |

| Model | Dataset | Math Avg@1[8] | Science Avg | Science Avg@1[4] | Average |
|---|---|---|---|---|---|
| $\mathcal{M}_{\text{base}}$ | - | 35.96 | 34.50 | 32.11 | 34.19 |
| | Only Math | 48.23 | 41.64 | 36.77 | 42.21 |
| $\mathcal{M}_{\text{RLP}}$ | Only Science | 49.17 | 39.65 | 38.26 | 42.36 |
| | Combined | 49.76 | 42.54 | 37.78 | 43.36 |

Table S.2: **Ablation on math, science, and combined domains.** RLP shows particularly strong generalization in presence of multi-domain data.

by summing the tokens used for gradient updates ($T_{inp}$) and the tokens processed during the rollout phase:

$$T_{flop} = (n \times l_{seq} \times bs \times iters) + T_{inp}$$

where $n$ is the number of rollouts per instance, $l_{seq}$ is the sequence length, $bs$ is the batch size and $iters$ is the number of steps RLP has gone through. This calculation results in an effective budget of approximately 6B tokens for our model. We therefore train a second, more powerful CPT baseline on 6B tokens (Base + CPT, Flop Usage), holding all other hyperparameters constant.

**RLP resonates well in presence of multidomain data.** Recent works have shown tremendous improvement in reasoning tasks, particularly in mathematics, through RLVR (Liu et al., 2025a; Luo et al., 2025; Hu et al., 2025). However, these methods are often tied to the complexity of queries, limiting their scalability. To draw a parallel, we evaluate RLP on Nemotron-Crossthink using different blends of math and science data. As shown in Table S.2, training only on math yields substantial math improvements, but comes at the cost of weaker generalization to science. Conversely, training only on science improves science accuracy, but underperforms in math compared to math-only training. Strikingly, combining both domains provides the best overall average, indicating that RLP is able to leverage complementary signals from multiple domains without diluting the benefits within each. This suggests that RLP not only scales beyond

| Benchmarks | $\mathcal{M}_{\text{base}}$ | $\mathcal{M}_{\text{RLP}}$ |
|---|---|---|
| MATH500 | 78.81 | 81.15 |
| GSM8K | 90.36 | 94.04 |
| AMC23 | 55.94 | 57.81 |
| Minerva | 37.96 | 40.26 |
| MMLU | 76.56 | 80.59 |
| MMLU@1[4] | 74.03 | 79.31 |
| MMLU-Pro | 59.20 | 65.53 |
| MMLU-Pro@1[4] | 54.00 | 61.75 |
| GPQA | 44.44 | 48.15 |
| GPQA@1[4] | 40.40 | 44.70 |
| Math Avg | 65.77 | 68.32 |
| Science Avg | 60.07 | 64.76 |
| Science Avg@1[4] | 56.14 | 61.92 |
| **Overall** | 60.66 | **65.00** |

Table S.3: RLP training with QWEN3-14B-BASE model.

single-domain specialization but also thrives in multidomain settings where diverse reasoning styles reinforce one another.

**Effectiveness of RLP with Scaling LLM.** Validating RLP on larger, state-of-the-art model sizes is indeed essential to confirm that our gains hold as parameter counts increase. To address this, we conducted a new set of experiments applying RLP to the $\mathcal{M} = $ QWEN3-14B-BASE model and training on our general pretraining corpus ($\mathcal{D}_{\mathrm{PT}}$) for 1B tokens. As shown in Table S.3, RLP delivers substantial improvements even on this stronger, significantly larger baseline. Applying RLP improves the overall average from 60.66% to 65.00%, with particularly notable gains in scientific reasoning where the average score improves from 60.07% to 64.76%. These results confirm that the dense, verifier-free signal provided by RLP remains effective at scale, successfully extracting reasoning capabilities that are not fully utilized by standard pretraining alone.

**How early RLP can be applied?** Previously, we have confirmed that RLP can be integrated to intermediate checkpoints from last stage of pretraining. However, it is unclear whether the gains sustain if we pick a very early checkpoint for RLP training. Inspired by the finding of Han et al. and to study how early RLP can be introduced, we additionally evaluate a much earlier checkpoint. Concretely, we take a NEMOTRON-NANO-12B-V2 model trained on only 4T tokens (about 20% of the full 20T pretraining budget) and apply RLP for 1B tokens on the same pretraining corpus $\mathcal{D}_{\mathrm{PT}}$. As shown in Table S.4, even at this early stage, RLP is highly effective: with only 1B tokens, Math Avg more than doubles (from 21.93 to 50.14), Science Avg@1[4] improves by 6 points (from 5.69 to 11.96), and Overall Average increases by 12 points (from 12.05 to 24.08). While our strongest final results come from applying RLP later in pretraining, these findings indicate that RLP can already yield large gains when the model has seen only a small fraction of the standard pretraining budget.

| Benchmarks | $\mathcal{M}_{\mathrm{base}}$ | $\mathcal{M}_{\mathrm{RLP}}$ |
|---|---|---|
| MATH500 | 30.15 | 62.38 |
| GSM8K | 29.56 | 81.42 |
| AMC23 | 22.81 | 37.81 |
| Minerva | 5.19 | 18.93 |
| MMLU | 11.59 | 13.10 |
| MMLU@1[4] | 8.73 | 20.68 |
| MMLU-Pro | 4.93 | 6.11 |
| MMLU-Pro@1[4] | 2.66 | 7.50 |
| GPQA | 9.10 | 11.20 |
| GPQA@1[4] | 5.68 | 7.70 |
| Math Avg | 21.93 | 50.14 |
| Science Avg | 8.54 | 10.14 |
| Science Avg@1[4] | 5.69 | 11.96 |
| **Overall** | 12.05 | 24.08 |

Table S.4: Comparison of NEMOTRON-NANO-12B-V2 4T Base and Base+RLP across benchmarks.

**FLOP matched comparison between RLP and RPT.** We would like to clarify that even though RLP can be theoretically applied to tokens at every position in the document, in practice we only apply it for one token per document. This token is selected randomly and not through any criteria as in the case of RPT. For the experiments in Table 3, we have matched the number of input tokens for both RLP and RPT settings (we train both methods for one epoch of the same documents). But we want to highlight that the number of target tokens for which reward is calculated is much larger for RPT compared to RLP (since we don't apply the RLP reward to every token in the document). Hence, the setting in Table 3 is in favor of RPT. Additionally, we don't include the compute needed to pre-select tokens using an external LLM for RPT.

To directly address the head to head flop matched comparison, we run a controlled experiment using Nemotron-CrossThink data. We deploy both RLP and RPT for only one epoch on the same data, i.e., the number of target tokens for which reward is calculated is similar in both cases. RLP achieves a 16.23% relative improvement in Overall Avg and consistently outperforms RPT on both math and science aggregates. These results confirm that the gains in Table 3 are not an artifact of mismatched settings. Even under stricter, target-matched conditions, RLP provides stronger and more general improvements.

As shown in Table 3, RLP achieves a 16.23% relative improvement in Overall Avg and consistently outperforms RPT on both math and science aggregates. These results confirm that the gains in Table 3 are not an artifact of mismatched settings. Even under stricter, target-matched conditions, RLP provides stronger and more general improvements.

**Continuous pretraining with longer context length.** Qwen3-1.7B-Base is indeed eventually extended to a 32K context window, but as described in the Qwen3 technical report, this happens only in a third long-context stage after the model has already been pretrained for 30T+ tokens at a much shorter context (4,096 tokens) and then further trained on knowledge-intensive data. Our CPT experiments are conceptually closer to these first two stages; we continue pretraining on our pretraining mixture ($\mathcal{D}_{\mathrm{PT}}$), which consists almost entirely of relatively short documents without long-range dependencies. In this regime, substantially increasing the context length does not obviously provide additional learning signal, but does change the optimization landscape and the effective batch and gradient statistics.

To evaluate the effect of longer context length, we conduct an additional controlled experiment where we keep all CPT hyperparameters fixed and only increased the context length from 8K to 32K. The resulting model, denoted ($\mathcal{M}_{\mathrm{CPT}}(32\mathrm{K})$), is compared to our original ($\mathcal{M}_{\mathrm{CPT}}(8\mathrm{K})$) in Table S.5.

The result suggests that for our pretraining corpus ($\mathcal{D}_{\mathrm{PT}}$), which rarely contains long documents that would actually utilize a 32K window, the 8K context configuration is at least as strong as, and in practice strictly better than, a 32K context configuration under matched compute and hyperparameters. Therefore, while we agree that context length is an important design choice, in our specific setup, using 8K rather than 32K does not weaken the CPT baseline; if anything, the longer context hurts optimization without yielding downstream benefits. Importantly, all comparisons between RLP and CPT are made against the stronger 8K CPT configuration.

| Benchmarks | $\mathcal{M}_{\mathrm{CPT}}(8\mathrm{K})$ | $\mathcal{M}_{\mathrm{CPT}}(32\mathrm{K})$ |
|---|---|---|
| AIME25 | 3.96 | 3.33 |
| MATH500 | 57.52 | 51.80 |
| GSM8K | 72.85 | 60.44 |
| AMC23 | 31.25 | 25.00 |
| Minerva | 19.03 | 17.46 |
| MMLU | 41.95 | 42.19 |
| MMLU@1[4] | 40.00 | 40.55 |
| MMLU-Pro | 27.81 | 27.08 |
| MMLU-Pro@1[4] | 24.61 | 22.87 |
| GPQA | 26.26 | 25.76 |
| GPQA@1[4] | 24.75 | 24.21 |
| Math Avg | 36.92 | 31.61 |
| Science Avg | 32.01 | 31.68 |
| Science Avg@1[4] | 29.79 | 29.21 |
| **Overall** | **32.90** | **30.83** |

Table S.5: Comparison of CPT models with 8K vs 32K context length.

**Effect of EMA over RLP training.** In RLP, the EMA baseline ($\bar{p}_\phi$) acts as a dynamic no-think counterfactual, providing a reference log-likelihood for each next token. The decay rate $\tau$ controls comparison difficulty: if $\tau$ is too low, the baseline updates too quickly and the reward collapses toward zero; if too high, it becomes stale and yields artificially easy gains.

To justify our choice of $\tau$, we ran a sensitivity study on Qwen3-1.7B-Base with $\tau \in 0.99, 0.995, 0.999, 0.9995$. As shown in Table S.6, performance forms a bell-shaped curve with a clear peak at $\tau = 0.999$.

| Model | $\tau$ | Math Avg | Science Avg | Overall Avg |
|---|---|---|---|---|
| $\mathcal{M}_{\mathrm{base}}$ | N/A | 35.96 | 32.11 | 34.03 |
| | 0.99 | 45.20 | 36.31 | 38.82 |
| | 0.995 | 45.18 | 37.36 | 39.21 |
| $\mathcal{M}_{\mathrm{RLP}}$ | **0.999** | **45.98** | **37.38** | **39.54** |
| | 0.9995 | 45.64 | 36.84 | 39.20 |

Table S.6: **Effect of temperature $\tau$ on performance.** Best result highlighted.

Across this range, training remained stable and we did not observe divergent or unstable behavior in any of our runs. Concerns that the model could "game" the objective by degrading the baseline do not manifest because the baseline is updated only via the EMA of the student parameters: for the baseline to degrade, the student must degrade first, which is immediately penalized through the primary reward term $\log p_\theta$. Thus the EMA baseline provides a stable, meaningful measure of information gain.

**Wall-clock time of RLP training versus SFT.** We conduct a direct comparison using 32 H100 GPUs with a global batch size of 512 and a 32K context length. As shown in the table below, RLP incurs an expected overhead due to the generation phase ($G = 16$ rollouts). While SFT, which has a similar computational profile to standard Continuous Pretraining (CPT), achieves a throughput of

92.34 samples/s (approx. 5.5s per step), RLP operates at 41.07 samples/s (approx. 12.5s per step). This results in a per-step slowdown factor of roughly $2.25\times$.

| Method | Batch Size | Rollouts ($G$) | Time/Step (s) | Throughput (samples/s) | Relative Speed |
|---|---|---|---|---|---|
| SFT | 512 | N/A | 5.54 | 92.34 | $1.00\times$ |
| RLP | 512 | 16 | 12.47 | 41.07 | $0.44\times$ |

Table S.7: Comparison of SFT and RLP training efficiency.

However, this per-step cost must be viewed in the context of convergence efficiency and total compute. While RLP is $2.25\times$ slower per iteration than SFT/CPT, it is drastically more data-efficient. As detailed in Table 4, RLP achieves an overall average accuracy of 43.36% on the Nemotron-Crossthink dataset using only 170M tokens. In contrast, the FLOP-matched CPT baseline required processing 6B tokens (roughly $35\times$ more data) to account for the compute difference, yet only reached an accuracy of 35.60%. Thus, while RLP processes tokens slower due to rollouts, the dense reward signal extracts significantly more reasoning capability per FLOP, yielding a performance margin (+7.76%) that standard training cannot replicate even with substantially higher data volume.

**Final perplexity after post-training.** We confirm that the model's perplexity on ordinary tokens does not degrade; in fact, it significantly improves. Unlike standard RLHF, where optimizing for an external reward often causes the model distribution to drift away from natural language, our reward signal is the log-likelihood of the next token itself. Therefore, by definition, RLP is optimizing for prediction accuracy.

| Model | Nemotron-CrossThink PPL ↓ | Nemotron-CrossThink NLL ↓ | Wikitext-103 PPL ↓ | Wikitext-103 NLL ↓ |
|---|---|---|---|---|
| $\mathcal{M}_{\text{base}}$ (**Qwen-1.7B**) | 2.91 | 1.06 | 5.83 | 1.77 |
| $\mathcal{M}_{\text{RLP}}$ (**Ours**) | **2.36** | **0.86** | **4.48** | **1.50** |

Table S.8: Perplexity and NLL comparison on Nemotron CrossThink and Wikitext-103.

Mathematically, maximizing the RLP reward is equivalent to minimizing the cross-entropy of the reasoned predictor against the data distribution (Proposition 1). As shown in the table below, our empirical results confirm this theoretical guarantee: $\mathrm{M_{RLP}}$ achieves consistently lower Perplexity (PPL) and Negative Log-Likelihood (NLL) compared to the base model. Crucially, this improvement holds for both the reasoning-intensive Nemotron CrossThink dataset and the general-domain Wikitext-103 benchmark, demonstrating that the "thoughts" generated by the model successfully compress information to better predict ordinary text.

**Computational Cost and FLOP Analysis of RLP.** A potential concern in comparing RLP against CPT is the perceived computational burden of autoregressively generating long reasoning traces. This would indeed be prohibitive if the rollout policy were applied at every token position in a sequence. In practice, however, RLP is applied to only *one randomly sampled token per sequence*, which dramatically reduces the computational burden. Instead of scaling with $L_{doc} \times L_{CoT}$, the rollout cost scales with $1 \times L_{CoT}$ per sequence. This design choice makes RLP computationally feasible and allows us to interleave it with standard training efficiently. In addition, autoregressive generation involves a bottleneck compared to parallel processing. We agree that this affects wall-clock time due to memory bandwidth constraints, but it does not incorrectly skew the *FLOP* calculation used for the baselines.

In Appendix 12, we calculated the FLOP-equivalent budget by summing the tokens used for gradient updates and the tokens generated during rollouts. We compared RLP (170M input tokens) against a CPT baseline trained on 6B tokens. This $35\times$ increase in data for the baseline is a rigorous upper bound for two reasons:

- **Operation Count:** The FLOPs of a forward pass for generating one token is approximately $2N$ (where $N$ is parameter count). The cost of training on one token (forward + backward) is approximately $6N$. By equating one generated token to one trained token in our FLOP calculation, we are effectively penalizing RLP (counting generation as 3x more expensive than it theoretically is in terms of FLOPs).

- **Total Compute:** Even with the overhead of 16 rollouts of length 2048 per document, the total floating-point operations performed by RLP on 170M documents are comparable to (or less than) performing standard forward/backward passes on the 6B tokens used in the $M_{CPT}$[6B] baseline.

While autoregressive generation is indeed slower in terms of wall-clock time, the purpose of the baseline is to compare compute efficiency. RLP applied to a single token per document is highly efficient, and our $M_{CPT}$[6B] baseline represents a compute-matched, which RLP still outperforms significantly (Overall Avg 42.13% vs 38.04%).

**On Self-Referentiality and the Semantics of the RLP Reward.** A natural concern for any method that leverages model-internal signals is whether the learning dynamics risk becoming self-referential—rewarding increases in internal confidence rather than genuine improvements in correctness or reasoning ability. In RLP, however, the reward structure is explicitly grounded in the data rather than in unconstrained model self-agreement.

- **Reward is anchored to ground-truth tokens.** For each sampled position, the reward

$$r(c_t) \;=\; \log p_\theta(x_t \mid x_{<t}, c_t) \;-\; \log \bar{p}_\phi(x_t \mid x_{<t})$$

  is defined with respect to the ground-truth next token $x_t$ from the corpus. Proposition 1 shows that, in expectation over $x_t \sim p^*(\cdot \mid x_{<t})$, this reward equals the reduction in cross-entropy achieved by conditioning on the thought $c_t$. A thought therefore receives positive reward only if it moves probability mass *toward* the correct continuation in the true data distribution. Increased confidence on an incorrect continuation strictly decreases the reward. This prevents the model from benefiting by simply inflating logit magnitudes or reinforcing patterns unrelated to accuracy.

- **EMA baseline prevents degenerate self-consistency loops.** RLP compares each thought-conditioned prediction to an exponential moving average (EMA) baseline $\bar{p}_\phi$ evaluated on the same context and same ground-truth token. If the current parameters shift toward patterns that improve internal consistency but harm prediction of the observed token, the relative likelihood under $p_\theta$ falls and the corresponding thought receives a negative advantage. Group-relative normalization and advantage clipping further ensure that thoughts cannot win reward by global logit scaling alone; only thoughts that contribute meaningful information about the next token outperform the EMA teacher in expectation.

- **External evaluations validate correctness rather than internal consistency.** The most important empirical question is whether internal information gain translates into better reasoning on verifiable tasks. Across GSM8K, MATH500, MMLU-Pro, GPQA, and other benchmarks with objectively checkable answers, RLP-trained models consistently outperform both the base model and compute-matched continuous-pretraining baselines—even when the latter consume substantially more training tokens at equal FLOPs. Notably, these gains persist after a strong post-training pipeline involving SFT and RLVR with external verifiers. If RLP were primarily amplifying internal confidence without improving correctness, these advantages would be expected to collapse or become fragile under verifier supervision. Instead, RLP-initialized models remain ahead, particularly on reasoning-heavy domains, indicating that the learned thoughts encode genuinely useful information and not merely self-reinforcing patterns.

Overall, the formulation of the RLP reward ensures that the model is optimized for meaningful reductions in predictive error on the underlying data distribution, while empirical evidence confirms that these internal information gains translate into improved external reasoning performance.

**Generalizability across Architectures and Data Distributions** To rigorously assess the universality of our approach, we evaluated RLP on two models chosen specifically for their significant divergence in both architectural design and data provenance: Qwen3-1.7B-Base and Nemotron-Nano-12B-V2. These distinct settings demonstrate that RLP is not limited to a single model family or training recipe.

- **Architectural Heterogeneity:** The models represent fundamentally different backbone architectures. Qwen3-1.7B-Base utilizes a standard, pure Transformer architecture. In contrast, Nemotron-Nano-12B-V2 is a *Hybrid Mamba2-Transformer*, which integrates State Space Models (SSM) with attention layers and employs a distinct tokenizer and training recipe. The fact that RLP translates

effectively to this hybrid architecture—driving an increase in overall average accuracy from 42.81% to 61.32%—provides strong evidence that the method is architecture-agnostic.

- **Data Lineage Independence:** The models also rely on distinct pretraining distributions. Nemotron-Nano-12B-V2 was trained from scratch on a massive mixture of raw public data (including Web Crawl, arXiv, and OpenWebMath) and synthetic data generated by a diverse ensemble of teacher models. As detailed in recent technical reports, this ensemble extends well beyond the Qwen family to include DeepSeek-R1 (DeepSeek-AI et al., 2025a), Mixtral-8x22B-v0.1 (Jiang et al., 2024), DeepSeek-V3 (DeepSeek-AI et al., 2025b), Phi-4 (Abdin et al., 2024), and Nemotron-4-340B (Nvidia et al., 2024).

By validating RLP on a Hybrid Mamba model trained on a multi-source ensemble distribution, distinct from the pure Transformer Qwen baseline, we confirm the method's scalability and robustness across disparate experimental environments.

**Analysis of thought traces.** We examine the model's internal thought traces to understand how RLP shapes reasoning. A manual analysis of 50 randomly sampled traces reveals consistently high-quality reasoning, characterized by grammatical correctness and strong contextual relevance, confirming that the information-gain objective effectively suppresses unproductive generation. We observe three qualitative properties: (1) *focused, context-aware reasoning*, where thoughts orient the model to the specific logical step required by the context; (2) *continuation-style* formatting, acting as internal scratch work that mirrors the prefix's tone and notation without meta-commentary; and (3) *utility-driven anticipation*, where thoughts predict structural moves (e.g., upcoming rules or algebraic steps) to maximize the likelihood of the ground-truth continuation. An illustrative example follows.

> **Prefix**
>
> "West Virginia students invited to enter Ornament Competition CHARLESTON, W.Va. (AP) — West Virginia students in kindergarten through 12th grade may enter the First Lady Student Ornament Competition this fall. First lady Cathy Justice is asking all students to participate in the 18th annual event. It is open to students in public and private schools as well as those who are home-schooled. Students are asked to create a "Nutcracker"-themed ornament for a tree to be displayed at the Culture Center in Charleston during the holidays, according to a news release from Gov. Jim Justice's office. There will be four divisions according to grade, and a winning class will be chosen from each division. The winning ornaments will be donated in January to the West Virginia State Museum. The four winning classes will"

> **Thought Trace**
>
> "The article is outlining contest logistics. The next sentence will likely add a specific detail such as what the winning classes receive, how the ornaments will be displayed, or other submission guidelines. Maintain the neutral news tone and extend the informational structure already established."

## 13 DATA BLEND EXTENDED RESULTS

To further examine whether RLP learns transferable reasoning beyond narrowly curated datasets, we evaluate it across a broad spectrum of corpora spanning both structured reasoning data and open-ended pretraining distributions. All experiments start from QWEN3-1.7B-BASE and apply RLP for 200 steps, consuming 170M input tokens, while keeping all other training settings fixed.

We consider two primary corpus families. The first consists of SFT-style reasoning datasets, including OmniMath (Gao et al., 2024), OpenThoughts (Guha et al., 2025), and Nemotron-CrossThink (Akter et al., 2025). These datasets contain structured question–answer pairs with explicit reasoning content and represent the typical setting where reinforcement-based methods are expected to perform well. The second family consists of general-purpose pretraining corpora, including academic papers (ACAD), math textbooks (Math-Text), and open-ended web crawl data. These datasets are not curated specifically for reasoning and more closely resemble large-scale pretraining mixtures.

| Benchmark | OmniMath | OpenThoughts | Nemotron-Crossthink | ACAD | Math-Text | Web-Crawl |
|---|---|---|---|---|---|---|
| MATH500 | 57.95 | 59.55 | 62.65 | 59.75 | 60.03 | 61.58 |
| GSM8K | 74.82 | 74.80 | 79.97 | 76.10 | 75.95 | 76.48 |
| AMC23 | 32.50 | 32.81 | 30.94 | 31.88 | 34.38 | 35.00 |
| Minerva | 20.63 | 20.96 | 25.46 | 22.98 | 21.92 | 22.43 |
| MMLU | 55.72 | 55.84 | 56.72 | 56.35 | 56.02 | 55.93 |
| MMLU@1[4] | 50.85 | 50.84 | 52.11 | 51.51 | 50.72 | 50.11 |
| MMLU-Pro | 35.81 | 34.55 | 38.58 | 35.11 | 36.06 | 37.02 |
| MMLU-Pro@1[4] | 31.47 | 30.57 | 35.10 | 31.06 | 31.86 | 32.43 |
| GPQA | 29.29 | 25.76 | 32.32 | 30.30 | 29.29 | 29.29 |
| GPQA@1[4] | 30.30 | 27.27 | 26.14 | 28.03 | 26.39 | 27.78 |
| Math Avg | 46.48 | 47.03 | 49.76 | 47.68 | 48.07 | 48.87 |
| Science Avg | 40.27 | 38.72 | 42.54 | 40.59 | 40.46 | 40.75 |
| Science Avg@1[4] | 37.54 | 36.23 | 37.78 | 36.87 | 36.32 | 36.77 |
| **Overall** | 41.43 | 40.66 | 43.36 | 41.71 | 41.62 | 42.13 |

Table S.9: Quantitative benchmarks for QWEN3-1.7B-BASE, showing the impact of RLP on different data blends. Shaded columns indicate general pretraining corpus.

Table 4 shows that RLP consistently improves over the base model across all corpus types. Importantly, the gains are not confined to math-centric SFT data. While the strongest improvements occur on Nemotron-CrossThink within the SFT family, substantial gains are also observed when training on purely general corpora such as academic papers and web crawl data. This demonstrates that RLP does not depend on carefully constructed reasoning traces. Instead, it extracts a transferable reasoning signal even from heterogeneous, weakly structured text.

A notable contrast emerges when compared with prior RL-based approaches that report improvements concentrated in high-quality math data and limited transfer to broader domains. In our experiments, models trained with mixed-domain or open-ended corpora simultaneously improve math, science, and professional benchmarks. There is no evidence of domain-specific overfitting or degradation on math when incorporating diverse data. Rather, diversity appears to strengthen general reasoning performance.

Table S.9 provides a task-level breakdown. Across MATH500, GSM8K, MMLU, MMLU-Pro, GPQA, and related metrics, improvements are observed regardless of whether the underlying training corpus is structured SFT data or general pretraining text. Even web-scale crawl data yields competitive math and science averages, suggesting that RLP leverages latent reasoning patterns embedded in broad distributions.

Overall, these results support three conclusions. First, RLP scales across corpus families without requiring specialized reasoning datasets. Second, it exhibits genuine cross-domain transfer rather than narrow task adaptation. Third, data diversity amplifies the learned reasoning signal instead of diluting it. Together, this positions RLP as a domain-agnostic pretraining augmentation that enhances both reasoning robustness and general benchmark accuracy.

