# OpenReview forum: "RLP: Reinforcement as a Pretraining Objective"
_ICLR.cc/2026/Conference — ICLR 2026 Poster_

### Official Review · Reviewer_cQyL · 2025-10-21

**Soundness:** 3
**Presentation:** 3
**Contribution:** 3
**Rating:** 6
**Confidence:** 4

**Summary:**

This paper introduces RLP, a pretraining paradigm that integrates RL into training reasoning LLMs. RLP reframes CoT generation as an action taken before next-token-prediction, and this action is rewarded by a verifier-free information gain which measures how much the CoT improves the model's likelihood of predicting the correct token. The no-think model adopts a DQN fashion update to keep its distribution close to the behavior policy. I think the authors' proposed method is a clever way to overcome the sparse reward and verifier-must conditions in applying RL to training a reasoning model.

**Strengths:**

* The paper presents a foundational shift by moving RL to the pretraining stage, and addresses the core limitation of standard next-token prediction.
* The information-gain reward is a key innovation, providing a dense, self-supervised signal on any text without needing external verifiers.

**Weaknesses:**

* While showing scalability, the most detailed experiments and ablations are on a relatively small 1.7B parameter model. The paper would be strengthened by a deeper quantitative analysis on a larger, more state-of-the-art model size (e.g., 7B+). Also, I would like to see the comparison with an instruct-tuned model.

**Questions:**

The no-think EMA baseline appears to serve a similar role to the target network in DQN, where the update frequency is a critical hyper-parameter that balances target stability against the staleness of the learning signal. In RLP, how sensitive are the training stability and final performance to the EMA decay rate? Did the authors experiment with different decay schedules for $\tau$, and can they characterize this trade-off?

---

> ### Author Response · Authors · 2025-11-25
> **Author Response to Reviewer cQyL - Part 1**
>
> Reviewer cQyL
>
> We thank the reviewer for the thoughtful and constructive comments. We found the suggestions especially helpful in strengthening the empirical scope and methodological clarity of the work, and we have conducted additional experiments and analyses directly addressing each point. Our detailed responses are provided below.
>
> > **The paper would be strengthened by a deeper quantitative analysis on a larger, more state-of-the-art model size (e.g., 7B+)**
>
>
>
> We appreciate the reviewer's emphasis on scaling analysis. Validating RLP on larger, state-of-the-art model sizes is indeed essential to confirm that our gains hold as parameter counts increase. To address this, we conducted a new set of experiments applying RLP to the Qwen3-14B-Base model and training on our general pretraining corpus for 1B tokens.
>
> As shown in the table below, RLP delivers substantial improvements even on this stronger, significantly larger baseline. Applying RLP **improves the overall average from 60.66% to 65.00%**, with particularly notable gains in scientific reasoning where the average score improves from **60.07% to 64.76%**. These results confirm that the dense, verifier-free signal provided by RLP remains effective at scale, successfully extracting reasoning capabilities that are not fully utilized by standard pretraining alone. We will incorporate these additional findings into the revised manuscript to further strengthen our empirical claims.
>
>
> | Model | math-500-pass@1[8] | gsm8k-pass@1[8] | amc23@1[8] | minerva_math-pass@1[8] | MMLU | MMLU-pass@1[4] | MMLU-Pro | MMLU-Pro-pass@1[4] | GPQA | GPQA-pass@1[4] | Math Avg | Science Avg | Science Avg-pass@1[4] | Overall Avg |
> | :--- | :--- | :--- | :--- | :--- | :--- | :--- | :--- | :--- | :--- | :--- | :--- | :--- | :--- | :--- |
> | Qwen3-14B-Base | 78.81 | 90.36 | 55.94 | 37.96 | 76.56 | 74.03 | 59.20 | 54 | 44.44 | 40.4 | 65.77 | 60.07 | 56.14 | 60.66 |
> | Qwen3-14B-Base + RLP | 81.15 | 94.04 | 57.81 | 40.26 | 80.59 | 79.31 | 65.53 | 61.75 | 48.15 | 44.7 | 68.32 | 64.76 | 61.92 | 65.00 |
>
> In addition, the paper already includes experiments on a larger 12B hybrid Mamba Transformer model (Nemotron Nano-V2-12B). In the original submission, we only reported improvement after applying RLP to the base model. To further address the reviewer's concern about scaling and to mirror the Qwen3 setting, we have now extended these experiments by applying the same two stage SFT + RLVR post training pipeline to both the Nemotron Nano-V2-12B base model and its RLP counterpart.
>
> The updated results as shown below confirms that RLP not only **yields a very large improvement at the base stage (Overall 42.81% to 61.32%, a 43% relative gain)** but that these gains persist and continue to compound after strong post training. After SFT + RLVR, the RLP trained model improves from 61.32% to 68.09%, maintaining a clear margin over the compute matched baseline (65.31%). The largest relative gains are in scientific reasoning as the Science Avg rises from 34.51% to 57.26% at the base stage and further to 64.52% after post training, compared to 58.91% for the continuously pretrained baseline. This pattern closely mirrors our findings on Qwen3, and demonstrates that RLP scales effectively both to larger parameter counts and to a different architecture family, while remaining compatible with strong downstream alignment. We will add these new results into the revised manuscript.
>
>
> | Benchmark        | $\mathcal{M}\_\mathrm{base}$ | $\mathcal{M}\_\mathrm{RLP}$  | $\mathcal{M}\_\mathrm{base}$ +Post | $\mathcal{M}\_\mathrm{RLP}$ +Post |
> |------------------|:------:|:-----:|:-----------:|:----------:|
> | MATH500          | 79.95  | 78.68 |    83.47    |   87.05    |
> | GSM8K            | 72.31  | 85.98 |    94.22    |   94.90    |
> | AMC23            | 70.63  | 57.19 |    62.19    |   75.00    |
> | Minerva          | 22.61  | 39.48 |    40.76    |   42.78    |
> | MMLU             | 54.12  | 78.76 |    73.55    |   78.17    |
> | MMLU@1[4]        | 48.01  | 79.48 |    75.23    |   77.90    |
> | MMLU-Pro         | 24.16  | 53.13 |    61.78    |   67.38    |
> | MMLU-Pro@1[4]    | 27.13  | 55.76 |    73.21    |   66.96    |
> | GPQA             | 25.25  | 39.90 |    41.41    |   48.00    |
> | GPQA@1[4]        | 22.47  | 48.86 |    52.15    |   49.62    |
> | Math Avg         | 61.38  | 65.33 |    70.16    |   74.93    |
> | Science Avg      | 34.51  | 57.26 |    58.91    |   64.52    |
> | Science Avg@1[4] | 32.54  | 61.37 |    66.86    |   64.83    |
> | **Overall**      | 42.81  | 61.32 |    65.31    | **68.09**  |

---

> ### Author Response · Authors · 2025-11-25
> **Author Response to Reviewer cQyL - Part 2**
>
> > **I would like to see the comparison with an instruct-tuned model.**
>
> We appreciate the reviewer’s interest in comparisons to instruct-tuned models and understand the motivation behind the request. At the same time, we are concerned that such a comparison would be difficult to interpret in the context of this work.
>
> Instruct-tuned checkpoints result from extensive post-training pipelines that use large instruction corpora, preference optimization, verifier or rater models, and multiple stages of RL. These pipelines vary widely across model families in data mixtures, filtering, and compute, most of which are not publicly documented. As a result, comparing RLP to an instruct-tuned model would mainly reflect differences in downstream data and alignment compute rather than the effect of the pretraining objective.
>
> RLP is fundamentally different. It is a pretraining-time objective applied directly to ordinary web and academic corpora without instruction labels, correctness signals, or curated reasoning traces. Instruct-tuned models rely on large post-training datasets that are not comparable to these domain-agnostic corpora, so head-to-head evaluations would not isolate the contribution of reinforcement-based pretraining.
>
> The central question of the paper is whether reinforcement can serve as a principled pretraining objective that improves the foundation model. To answer this, we evaluate in a controlled setting where all models share the same architecture and downstream post-training, differing only in the pretraining objective. In this matched setup, RLP consistently improves both the base model and the aligned model, showing that the gains arise from reinforcement at pretraining time and persist after alignment.
>
> For these reasons, we focus on controlled comparisons that isolate RLP’s effect, since adding instruct-tuned baselines would obscure the signal by introducing unrelated differences from heterogeneous post-training pipelines.
>
> > **In RLP, how sensitive are the training stability and final performance to the EMA decay rate? Did the authors experiment with different decay schedules for and can they characterize this trade-off?**
>
> We appreciate the insightful comparison to DQN target networks, which accurately captures the dynamic role of the EMA baseline in RLP. In our framework, the decay rate $\tau$ regulates the "difficulty" of the reward signal. The reward is the log-likelihood ratio between the reasoning policy and the no-think baseline. If $\tau$ is too low (fast update), the baseline tracks the current model too closely, causing the reward signal to collapse toward zero and reducing the incentive to explore complex thoughts. Conversely, if $\tau$ is too high (slow update), the baseline becomes stale, creating an artificially easy comparison that may not reflect true information gain relative to the model's current non-reasoning capabilities. Our choice of $\tau=0.999$ was designed to strike a balance, maintaining a baseline that is sufficiently lagged to provide a meaningful contrast, yet fresh enough to stabilize the variance of the advantage estimator.
>
> To empirically characterize this trade-off, we conducted a sensitivity sweep with $\tau \in \{0.99, 0.995, 0.999, 0.9995\}$ on the Qwen3-1.7B-Base model. As shown in the table below, performance follows a bell-shaped curve peaking at our chosen value of $\tau=0.999$ (Overall Avg 39.54\%). Reducing the decay to $\tau=0.99$ drops the Overall Average to 38.82\%, consistent with the hypothesis that a fast-moving baseline dampens the learning signal. Similarly, increasing $\tau$ to 0.9995 leads to a slight regression (39.20\%), suggesting that an excessively stale baseline yields diminishing returns. While RLP remains relatively robust across this range, maintaining strong improvements over the base model in all configurations, the data confirms that $0.999$ provides the optimal balance for maximizing reasoning performance across both math and science domains.
>
> | Model | $\tau$ | MATH-500 | GSM8K | AMC23 | Minerva | MMLU | MMLU-Pro | GPQA | Math Avg | Science Avg | Overall Avg |
> | :--- | :--- | :--- | :--- | :--- | :--- | :--- | :--- | :--- | :--- | :--- | :--- |
> | Qwen3-1.7B-Base | 0.99 | 58.35 | 73.93 | 28.12 | 20.45 | 54.22 | 30.98 | 23.74 | 45.20 | 36.31 | 38.82 |
> | Qwen3-1.7B-Base | 0.995 | 58.17 | 73.26 | 29.06 | 20.22 | 54.29 | 31.03 | 26.77 | 45.18 | 37.36 | 39.21 |
> | **Qwen3-1.7B-Base** | **0.999** | **58.78** | **73.64** | **31.00** | **20.50** | **54.43** | **31.00** | **26.70** | **45.98** | **37.38** | **39.54** |
> | Qwen3-1.7B-Base | 0.9995 | 57.38 | 73.80 | 30.94 | 20.45 | 54.42 | 30.85 | 25.25 | 45.64 | 36.84 | 39.20 |
>
> We will add these findings to the revised manuscript of our submission.

---

> > ### Comment · Reviewer_cQyL · 2025-11-28
> >
> > Thank you for the responses, they addressed my concerns and I'll keep my positive rating.

---

### Official Review · Reviewer_kYzE · 2025-10-24

**Soundness:** 1
**Presentation:** 3
**Contribution:** 2
**Rating:** 2
**Confidence:** 3

**Summary:**

This work introduces a new method for applying RL right after the pre-training phase. In particular, this involves generating reasoning traces with a base model and computing a reward signal based on how well these reasoning traces increase the likelihood of ground-truth next-tokens labels in a given dataset. The authors call this method RLP and test it on top of a Qwen 1.7B base transformer model and a Nemotron hybrid model, evaluating directly on generation-based tasks.

**Strengths:**

- Overall, the exposition of the method is clear, and the simple theoretical considerations based on Jensen's inequality in section 2.2 to show the loss is a sensible optimization objective are concise and not overemphasized.
- The authors apply their method to a hybrid SSM like Nemotron. Moreover, they also experiment with different data sources in Table 4. This degree of diversity is nice to see.
- While code is not released, the authors do provide training and hyper-parameter details in Appendices 9 and 10, which provide helpful transparency to the method and baselines considered.

**Weaknesses:**

**Main concerns**
My main concerns are about the soundness of the paper's empirical validation. The main quantitative results in tables 1, 3, and 4 feature a Qwen3 1.7B base model. This base model is extended and compared with RLP on downstream reasoning tasks. There are several aspects about how the evaluation is carried out and the baselines that prompted my concerns:
1) Evaluating base models without any alignment on generation tasks and without even "few-shot" examples is doomed to underperform, as these models have never been primed for question-answering. Thus, I am unsure how relevant it would be to compare RLP with continued pretraining and base models trained on fixed non-masked datasets (which make up the bulk of the reported results), as pretraining corpora are purposefully not meant to imbue the models with alignment capabilities. The RLP traces generated with low temperature (appendix 9) and filtered based on likelihood should inherently make the model's logits more skewed (something that even occurs in RL with random rewards [1]).
2) Looking at the hyperparameters for the continued pre-training baseline (CPT), these appear quite odd and suboptimal (e.g., using a much smaller context than what Qwen3 was pretrained with in the first place). Beyond pretraining, the "post-training" procedure in Table 1 uses the OpenThought dataset that was conceived for finetuning already post-trained "instruct" models, and is much smaller and less diverse than common datasets used for the first SFT phase.
3) One of the key downsides of RL is its disproportionate cost compared to other training stages. Yet, with one exception, all "continued pre-training baselines" seem to only equate the size of RLP's "seed" dataset, which I believe implies that the training budget and tokens used for CPT would be disproportionately more (likely well over an order of magnitude). There is a single baseline in Table 4 that is described to be "equating FLOPs", yet, looking at Section 10 in the Appendix, this FLOP calculation does not take into account the autoregressive bottleneck of generating reasoning traces with RL. If I am not missing something, I believe this likely implies the training time for CPT would still be much larger.

Based on 1), 2), and 3), I find it hard to extrapolate that the proposed method would provide any benefit to an actual LLM training pipeline and would be worth the high computational demands of autoregressive generation at an early training stage. In order to provide convincing evidence, I would suggest comparing their final model directly to the "instruct" version of Qwen 3 1.7B, but I suspect that given 2), even their final post-trained version would significantly underperform with respect to this properly aligned baseline.

**Other**
1. I did not find the role of the EMA model to compute the baseline to be properly explained. While the authors briefly mention the reward hacking hypothesis, this does not seem to be empirically validated. Empirically, did the authors find divergent behaviors without EMA? Keeping EMA in memory also requires storing a separate copy of the network's weight, with immediate additional cost concerns that are not currently discussed.
2. When comparing with RPT, the authors mention 1 epoch training "with matched data and compute". Yet, it's unclear to me how both can be true at the same time, as RPT performs token-level filtering before training. To equate compute, RLP would have to train on a much smaller number of unfiltered datapoints from Omni-MATH.
3. Without strong empirical validation or analysis, the overall contribution of the paper seems quite limited to me. Generating reasoning traces during pretraining was already done by [2] (which the authors do cite), while rewarding reasoning traces with the log probabilities of ground-truth completion tokens was already done by [3] (currently, not cited).


[1]  Shao, Rulin, et al. "Spurious rewards: Rethinking training signals in rlvr." arXiv preprint arXiv:2506.10947 (2025).

[2] Hatamizadeh, Ali, et al. "RLP: Reinforcement as a Pretraining Objective." arXiv preprint arXiv:2510.01265 (2025).

[3] Cetin, Edoardo, Tianyu Zhao, and Yujin Tang. "Reinforcement Learning Teachers of Test Time Scaling." arXiv preprint arXiv:2506.08388 (2025).

**Questions:**

I have detailed what I found to be the biggest areas of improvement and my questions about this work in the Weaknesses section of my review. I would encourage the authors to address these points and help clarify any potential misconceptions for the discussion phase.

---

> ### Public Comment · ~Chaojian_Shi1 · 2025-11-14
>
> Hello!
> In the weakness section, the reviewer wrote:
>
> “Without strong empirical validation or analysis, the overall contribution of the paper seems quite limited to me. Generating reasoning traces during pretraining was already done by [2] (which the authors do cite), while rewarding reasoning traces with the log probabilities of ground-truth completion tokens was already done by [3] (currently, not cited).”
>
> However, reference [2] (Hatamizadeh et al., "RLP: Reinforcement as a Pretraining Objective," arXiv:2510.01265, 2025) is actually the very paper under review.
> So it seems the reviewer is inadvertently citing this same submission as prior work.

---

> ### Author Response · Authors · 2025-11-25
> **Clarifications on Key Factual Inaccuracies in the Review**
>
> We thank Reviewer kYzE for their time and feedback. Before providing our detailed point-by-point response, we want to clearly correct several factual inaccuracies that significantly shaped the review.
>
> ### **Factual Inaccuracy 1: No novelty over a future paper**
>
> The review states that our contribution lacks novelty because *"Generating reasoning traces during pretraining was already done by [2]."*
>
> **This is incorrect.** Reference [2] is not prior work. As the external commenter pointed out, it has the same title as our submission, and appeared on arXiv AFTER the ICLR deadline. Thus, it was not (and couldn't possibly be) cited in our anonymized submission and cannot be used to argue that our work is derivative. Treating a post-deadline upload as prior art fundamentally misrepresents our novelty.
>
> ### **Factual Inaccuracy 2: Context window size**
>
> The review describes our CPT hyperparameters as "odd and suboptimal," claiming we used *"a much smaller context than Qwen3."*
>
> **This is not accurate.** Qwen3-Base was pretrained with a 4k context, which was only extended during post-training [(Qwen3 technical report)](https://arxiv.org/pdf/2505.09388). Our CPT setup uses an 8k window, which is larger than the original pretraining context. The claim that we used a smaller or suboptimal context window is factually false.
>
> ### **Factual Inaccuracy 3: RPT comparison and compute**
>
> The review questions how our comparison with RPT could match "data and compute," given RPT's token filtering.
>
> **The comparison was fully controlled.** RPT and RLP were trained on the same number of input tokens, and RPT actually computes reward on more target tokens than RLP, which biases the setup in RPT's favor. The auxiliary filtering model does not alter the matched token budget used for the comparison.
>
> ### **Factual Inaccuracy 4: FLOPs calculation**
>
> The review states that our FLOP-equating baseline *"does not include the autoregressive bottleneck."*
>
> **This is contradicted by our paper.** Section 11 of the Appendix explicitly shows that our FLOP calculation fully includes the autoregressive rollout cost, counting all tokens processed during decoding. The claim that this overhead was ignored is inaccurate.
>
> We respectfully ask the reviewer to reassess their evaluation in light of these clarifications. A complete response to the remaining qualitative points will follow shortly.

---

> ### Comment · Reviewer_kYzE · 2025-11-25
> **Thanks for your rebuttal and pointing out my reference mistake**
>
> First and foremost, I would like to **apologize for citing the wrong reference** and thank both the authors and the public commenter for kindly pointing out my mistake.
>
> The correct citation is to RPT [2], but, unfortunately, as I searched for the correct reference "Reinforcement Pre-Training," the concurrent arXiv submission with the same name as this submission came up as the second result, and I mistakenly picked it up. I have now edited my review to fix this, in order to limit making other reviewers aware of this concurrent open submission.
>
> Regardless, though, I would kindly ask the authors to avoid mischaracterizing my review and reconsider some of the main concerns highlighted. Below, I address their current rebuttal points:
>
> 1) In my review, I never stated "No novelty". The **last of my non-major concerns** was that the contribution of this work seemed quite incremental, given the very similar methodology to RPT [2], which the authors did cite predominantly in their submission. While I believe incremental work can be very valuable, this needs to be backed by strong empirical gains with a sensible training and evaluation setup. Once again, as this was the last of the non-major concerns in my review, I would encourage the authors to first prioritize the other major suggested areas of potential improvements, which I believe at the time of submission were actionable weaknesses.
>
> 2) Qwen3 Base **has a context of 32768, not 4096**. This is clearly stated both in the model card (see https://huggingface.co/Qwen/Qwen3-1.7B-Base) as well as the Qwen 3 paper, which the authors even linked in their rebuttal, e.g., from the 4th paragraph of the introduction:
>
> "The pre-training process follows a three-stage strategy. In the first stage, the model is trained on about 30 trillion tokens to build a strong foundation of general knowledge. In the second stage, it is further trained on knowledge-intensive data to enhance reasoning abilities in areas like science, technology, engineering, and mathematics (STEM) and coding. Finally, in the third stage, the model is trained on long-context data to increase its maximum context length from 4,096 to 32,768 tokens."
>
> 3) The statement that "RPT and RLP were trained on the same number of input tokens, and RPT actually computes reward on more target tokens than RLP" was never mentioned in the paper and seems to contradict the previous wording:
>
> "both methods for one epoch under matched data and compute" (line 357-359)
>
> If RPT were also to be trained for 1 epoch, it would actually optimize for far fewer tokens, given its auxiliary filtering mechanism. Thus, I would kindly ask the authors to clarify in their paper and in their following responses if they used some additional unspecified data source, or for how many epochs RPT was actually trained.
>
>
> 4) From Section 11 of the submitted paper, the equation for FLOP computation is simply adding the total generated tokens to the input tokens:
>
> $\( T_{\text{flop}} = (n \times l_{\text{seq}} \times bs \times iters) + T_{\text{inp}} \)$
>
> As detailed in my review, while many of the baselines do not equate FLOPs, even the ones that are, they do not seem to equate actual training costs. This is because the new tokens need to be **autoregressively generated**: for every sequence of l_seq length, this requires l_seq additional forward passes instead of one. Thus, unless I am misunderstanding some crucial about the method, the training time (including autoregressive generation) should be considerably larger.
>
> [2] Dong, Qingxiu, et al. "Reinforcement Pre-Training." arXiv preprint arXiv:2506.08007 (2025).

---

> > ### Author Response · Authors · 2025-11-25
> > **Thank you for rapid engagement and quick response**
> >
> > Thanks much for clarifying that the novelty was only a minor concern! Nonetheless, in case the relative novelty wasn't clear due to the similar paper titles, we'd like to point out the significant contributions our paper uniquely adds over [2]:
> >
> > - Unlike RPT, RLP introduces a verifier-free information gain objective that augments next-token prediction by rewarding thoughts in proportion to their predictive utility (explained in Line 052-077).
> > - RLP does not rely on an external LLM to identify target tokens, which makes it straightforward to integrate directly into pretraining pipelines (Line 075-077).
> > - RLP can be theoretically applied to every token in a document whereas RPT can only be applied to select target tokens (Line 076-078).
> > - RPT has demonstrated their technique only on distilled checkpoints whereas RLP has been applied to intermediate pretraining checkpoints of Nemotron-Nano-12B-V2 (Line 078-079 and Table 2).
> >
> > Hence, we respectfully disagree that RLP is fundamentally “very similar methodology to RPT”.
> >
> > We very much appreciate the reviewer's clarification regarding Qwen3’s context window. Our earlier statement referred specifically to the base general pretraining phase, which uses a 4k window. Given the clarification of the reviewer, we now fully understand the basis of the reviewer's concern and will provide a detailed clarification in our updated response.
> >
> > Regarding the RPT and FLOP computation, we realize that there is a misconception related to how we train RLP. Although RLP can theoretically compute rewards at every token position in a document, in practice, **we only apply RLP to one randomly sampled token per document**. This is more token efficient than the RPT experiment we have in the paper, as for RPT, there can be multiple high-entropy tokens within a single document. Hence, in Table 3 RPT reward is applied on more target tokens compared to RLP. We hope this clarifies the reviewer’s concerns about RPT and FLOP computation.

---

> > > ### Author Response · Authors · 2025-11-25
> > > **Further Evidence on RPT comparison and FLOP matched setting**
> > >
> > > To reiterate, for the experiments in Table 3, we have matched the number of input tokens for both RLP and RPT settings (we train both methods for one epoch of the same documents). But we want to highlight that **the number of target tokens for which reward is calculated is much larger for RPT compared to RLP** (since we don’t apply the RLP reward to every token in the document). Hence, the setting in Table 3 is in favor of RPT. Additionally, we don’t include the compute needed to pre-select tokens using an external LLM for RPT.
> > >
> > > To directly address the reviewer’s concern, we ran a controlled experiment using Nemotron-CrossThink data. We deploy both RLP and RPT for only one epoch of Nemotron-CrossThink data, i.e., **the number of target tokens for which reward is calculated is same in both cases**. It is matched in terms of  number of input documents and the data itself, as well as matched in the number of target tokens used to calculate reward, and finally also matched in the total FLOPs used because we use same parameters for both the experiments such as sequence length, number of rollouts etc.
> > >
> > >
> > > | Model     | math-500-pass@1[8] | gsm8k-pass@1[8] | amc23@1[8] | minerva_math-pass@1[8] | MMLU  | MMLU-pass@1[4] | MMLU-Pro | MMLU-Pro-pass@1[4] | GPQA  | GPQA-pass@1[4] | Math Avg | Science Avg | Science Avg-pass@1[4] | Overall Avg |
> > > |-----------|--------------------|-----------------|-------------|-------------------------|-------|----------------|-----------|---------------------|--------|----------------|-----------|--------------|------------------------|--------------|
> > > | RPT [NC]  | 49.03              | 55.71           | 25.31       | 16.59                  | 47.61 | 45.292         | 27.75    | 25.07               | 27.78 | 26.01          | 36.66    | 34.38        | 32.12                 | 34.39       |
> > > | RLP [NC]  | 58.33              | 73.77           | 31.25       | 20.45                  | 54.06 | 50.45          | 31.92    | 28.63               | 30.3  | 26.52          | 45.95    | 38.76        | 35.20                 | 39.97       |
> > >
> > >
> > > As shown in the results above, **RLP achieves a 16.23% relative improvement** in Overall Avg and consistently outperforms RPT on both math and science aggregates. These results confirm that the gains in Table 3 are not an artifact of mismatched settings. Even under stricter, target-matched conditions, RLP provides stronger and more general improvements. We highly appreciate the concern raised by the reviewer as the head to head comparison further strengthens our claim. We will incorporate the results in our final draft.

---

> ### Author Response · Authors · 2025-11-27
> **Final Author Response to Reviewer kYzE - Part 1**
>
> We thank the reviewer for their time and effort in reviewing our manuscript. We provide detailed responses below.
>
> > **My main concerns are about the soundness of the paper's empirical validation. The main quantitative results in tables 1, 3, and 4 feature a Qwen3 1.7B base model. This base model is extended and compared with RLP on downstream reasoning tasks.**
>
> Our validation **extends beyond Qwen3-1.7B**. While the 1.7B model enabled efficient ablations, Section 4 shows that RLP **scales effectively** to the much larger **Nemotron-Nano-12B-V2**, a hybrid Mamba–Transformer with **about seven times more parameters** and a different architecture.
>
> On the 12B base model, RLP raises overall accuracy from 42.81% to 61.32%, exceeding the 19% improvement observed on Qwen3-1.7B, and delivers a 23% absolute gain in Scientific Reasoning. These results confirm that RLP’s benefits **grow with model size** and transfer robustly across architectures.
>
> To further address scaling and match the Qwen3 pipeline, we also applied the same SFT and RLVR post-training procedure to both the **base** and **RLP-pretrained** 12B models. As shown in the table below, RLP not only yields large improvements on the base model, but these gains **persist and continue to accumulate** after post-training: the RLP model reaches **68.09%**, compared with 65.31% for the compute-matched baseline, with especially strong effects in scientific reasoning (64.52%).
>
> | Benchmark        | $\mathcal{M}\_\mathrm{base}$ | $\mathcal{M}\_\mathrm{RLP}$  | $\mathcal{M}\_\mathrm{base}$ +Post | $\mathcal{M}\_\mathrm{RLP}$ +Post |
> |------------------|:------:|:-----:|:-----------:|:----------:|
> | MATH500          | 79.95  | 78.68 |    83.47    |   87.05    |
> | GSM8K            | 72.31  | 85.98 |    94.22    |   94.90    |
> | AMC23            | 70.63  | 57.19 |    62.19    |   75.00    |
> | Minerva          | 22.61  | 39.48 |    40.76    |   42.78    |
> | MMLU             | 54.12  | 78.76 |    73.55    |   78.17    |
> | MMLU@1[4]        | 48.01  | 79.48 |    75.23    |   77.90    |
> | MMLU-Pro         | 24.16  | 53.13 |    61.78    |   67.38    |
> | MMLU-Pro@1[4]    | 27.13  | 55.76 |    73.21    |   66.96    |
> | GPQA             | 25.25  | 39.90 |    41.41    |   48.00    |
> | GPQA@1[4]        | 22.47  | 48.86 |    52.15    |   49.62    |
> | Math Avg         | 61.38  | 65.33 |    70.16    |   74.93    |
> | Science Avg      | 34.51  | 57.26 |    58.91    |   64.52    |
> | Science Avg@1[4] | 32.54  | 61.37 |    66.86    |   64.83    |
> | **Overall**      | 42.81  | 61.32 |    65.31    | **68.09**  |
>
> > **Evaluating base models without any alignment on generation tasks and without even "few-shot" examples is doomed to underperform, as these models have never been primed for question-answering.**
>
> Our design **controls for alignment and prompting effects** by reporting base model metrics only to isolate what pretraining imparts, while drawing conclusions from performance **after identical post-training (SFT + RLVR)**, where all models share the same alignment and QA-ready prompting.
>
> As shown in the **“Post” columns of Table 1**, the advantage of **RLP** persists and even **strengthens** after matched post-training. The post-trained RLP model exceeds the baseline by **8% on average**, including clear gains on reasoning benchmarks such as **MMLU-Pro (42.40 vs 37.85)**. With architecture, data, post-training, and evaluation held constant, these comparisons cleanly **isolate the effect of the pretraining objective**.
>
> Base-model evaluations are not meant to maximize absolute accuracy, but to **probe what pretraining alone provides**. All models use the same zero-shot setup and decoding, so any absolute underperformance is shared and does not affect **relative** differences. The consistent RLP gains show improved internal reasoning before alignment.
>
> > **I am unsure how relevant it would be to compare RLP with continued pretraining and base models trained on fixed non-masked datasets**
>
> We believe comparing RLP to continuous pretraining (CPT) and base models is not only relevant but essential for evaluating RLP as a **pretraining objective**:
>
> 1. Because RLP is intended to replace or augment next-token prediction, the correct control is a model trained with the standard pretraining objective on identical data. Comparing RLP to CPT isolates the effect of the **information-gain reward**, distinguishing improvements due to the method rather than the dataset.
>
> 2. While pretraining corpora are not for alignment, our results show that RLP yields a stronger *reasoning foundation* than standard pretraining. To ensure this is not an alignment artifact, all models undergo the **same post-training pipeline (SFT + RLVR)**.
>
>    * If RLP merely acted like alignment, its advantages would disappear after post-training.
>    * Instead, they **grow**: the post-trained RLP model beats the post-trained CPT model by **7–8%**.
>
> Thus the base-model comparison is appropriate and meaningful.

---

> ### Author Response · Authors · 2025-11-27
> **Final Author Response to Reviewer kYzE - Part 2**
>
> > **The RLP traces generated with low temperature and filtered based on likelihood should inherently make the model's logits more skewed**
>
> In **RLP**, the only place where log likelihood enters the update is through the information-gain reward
>
> $$
> r(c_t) = \log p_\theta(x_t \mid x_{<t}, c_t);-;\log \bar p_\phi(x_t \mid x_{<t}),
> $$
>
> and we apply GRPO-style updates only on the thought tokens. All (G) sampled thoughts for a given context—including low-likelihood ones—are used to form the group relative advantages; we do **not** resample or accept/reject trajectories based on likelihood. The rollout temperature in Appendix 9 is chosen only to keep thoughts coherent and does not alter the reward signal, which is defined relative to the EMA baseline. Thus, a uniform sharpening of logits that does **not** improve next-token prediction receives no positive advantage and is not reinforced.
>
> Regarding the connection to RL with *random rewards*, note that with our group-relative baseline
>
> $$
> A^{(i)} = \frac{G}{G-1}\bigl(r^{(i)} - \bar r\bigr),
> $$
>
> any reward independent of the sampled thought has zero expected advantage, producing a vanishing policy gradient. **Proposition 1** shows that the expected reward at each position equals the reduction in cross-entropy of the thought-conditioned predictor relative to the EMA baseline, and **Proposition 3** extends this to sequences. Consequently, unlike setups where a fixed positive reward can globally skew logits, **RLP** can only push the model toward thoughts that genuinely increase the probability of observed tokens under the data distribution.
>
> > **Smaller context length for Qwen3 CPT.**
>
> Qwen3-1.7B-Base is indeed eventually extended to a 32K context window, but as described in the Qwen3 technical report, this happens only in a *third* long-context stage after the model has already been pretrained for 30T+ tokens at a much shorter context (4,096 tokens) and then further trained on knowledge-intensive data. Our CPT experiments are conceptually closer to these first two stages; we continue pretraining on our pretraining mixture (\mathcal{D}_{\mathrm{PT}}), which consists almost entirely of relatively short documents without long-range dependencies. In this regime, substantially increasing the context length does not obviously provide additional learning signal, but does change the optimization landscape and the effective batch and gradient statistics.
>
> To directly address the reviewer's concern, we ran an additional controlled experiment where we kept all CPT hyperparameters fixed and only increased the context length from 8K to 32K. The resulting model, denoted (M_{\mathrm{CPT}}(32\mathrm{K})), is compared to our original (M_{\mathrm{CPT}}(8\mathrm{K})) in the table below.
>
> | Benchmark        | M_CPT(8K) | M_CPT(32K) |
> | ---------------- | --------- | ---------- |
> | AIME25           | 3.96      | 3.33       |
> | MATH500          | 57.52     | 51.80      |
> | GSM8K            | 72.85     | 60.44      |
> | AMC23            | 31.25     | 25.00      |
> | Minerva          | 19.03     | 17.46      |
> | MMLU             | 41.95     | 42.19      |
> | MMLU@1[4]        | 40.00     | 40.55      |
> | MMLU-Pro         | 27.81     | 27.08      |
> | MMLU-Pro@1[4]    | 24.61     | 22.87      |
> | GPQA             | 26.26     | 25.76      |
> | GPQA@1[4]        | 24.75     | 24.21      |
> | Math Avg         | 36.92     | 31.61      |
> | Science Avg      | 32.01     | 31.68      |
> | Science Avg@1[4] | 29.79     | 29.21      |
> | Overall          | 32.90     | 30.83      |
>
> These results suggest that for our pretraining corpus (\mathcal{D}_{\mathrm{PT}}), which rarely contains long documents that would actually utilize a 32K window, the 8K context configuration is at least as strong as, and in practice strictly better than, a 32K context configuration under matched compute and hyperparameters.
>
> Therefore, while we agree that context length is an important design choice, in our specific setup, using 8K rather than 32K does not weaken the CPT baseline; if anything, the longer context hurts optimization without yielding downstream benefits. Importantly, all comparisons between RLP and CPT are made against the stronger 8K CPT configuration.

---

> ### Author Response · Authors · 2025-11-27
> **Final Author Response to Reviewer kYzE - Part 3**
>
> > **The FLOP calculation does not take into account the autoregressive bottleneck of generating reasoning traces with RL.**
>
> The reviewer's concern stems from a valid intuition. If RLP were applied to *every* token position in a document (generating a chain-of-thought for every token $x_t$ in the document), the computational cost would indeed be astronomically higher than standard pretraining. However, we wish to clarify that **we apply RLP to only one randomly sampled token per sequence during training.**
>
> While the reward signal is *available* at every position theoretically, computing it sparsely is sufficient for the model to learn the reasoning policy. This dramatically reduces the computational burden. Instead of scaling with $L_{doc} \times L_{CoT}$, the rollout cost scales with $1 \times L_{CoT}$ per sequence. This design choice makes RLP computationally feasible and allows us to interleave it with standard training efficiently.
>
> The reviewer also notes that autoregressive generation involves a bottleneck compared to parallel processing. We agree that this affects wall-clock time due to memory bandwidth constraints, but it does not incorrectly skew the *FLOP* calculation used for the baselines.
>
> In the Appendix, we calculated the FLOP-equivalent budget by summing the tokens used for gradient updates and the tokens generated during rollouts. We compared RLP (170M input tokens) against a CPT baseline trained on 6B tokens. This 35$\times$ increase in data for the baseline is a rigorous upper bound for two reasons:
>
> * **Operation Count:** The FLOPs of a forward pass for generating one token is approximately $2N$ (where $N$ is parameter count). The cost of training on one token (forward + backward) is approximately $6N$. By equating one generated token to one trained token in our FLOP calculation, **we are effectively penalizing RLP** (counting generation as 3x more expensive than it theoretically is in terms of FLOPs)
>
> * **Total Compute:** Even with the overhead of 16 rollouts of length 2048 per document, the total floating-point operations performed by RLP on 170M documents are comparable to (or less than) performing standard forward/backward passes on the 6B tokens used in the $\mathrm{M}_{\mathrm{CPT}}$[6B] baseline.
>
> While autoregressive generation is indeed slower in terms of wall-clock time, the purpose of the baseline is to compare *compute efficiency*. RLP applied to a single token per document is highly efficient, and our $\mathrm{M}_{\mathrm{CPT}}$[6B] baseline represents a compute-matched, which RLP still outperforms significantly (Overall Avg 42.13\% vs 38.04\%).
>
> We will add a discussion in the revised manuscript clarifying why applying RLP to a single sampled token per sequence makes the rollout compute comparable to the CPT-6B baseline, addressing the reviewer’s concern regarding autoregressive bottlenecks and FLOP accounting.
>
> > **I did not find the role of the EMA model to compute the baseline to be properly explained. Did the authors find divergent behaviors without EMA?**
>
> In RLP, the EMA baseline ($\bar p_\phi$) acts as a **dynamic no-think counterfactual**, providing a reference log-likelihood for each next token. The decay rate $\tau$ controls comparison difficulty: if $\tau$ is too low, the baseline updates too quickly and the reward collapses toward zero; if too high, it becomes stale and yields artificially easy gains.
>
> To justify our choice of $\tau$, we ran a sensitivity study on Qwen3-1.7B-Base with $\tau \in {0.99, 0.995, 0.999, 0.9995}$. As shown below, performance forms a bell-shaped curve with a clear peak at **$\tau=0.999$**.
>
> | Model     | $\tau$    | Math Avg  | Science Avg | Overall Avg |
> | :-------- | :-------- | :-------- | :---------- | :---------- |
> | Reference | N/A       | 35.96     | 32.11       | 34.03       |
> | RLP       | 0.99      | 45.20     | 36.31       | 38.82       |
> | RLP       | 0.995     | 45.18     | 37.36       | 39.21       |
> | **RLP**   | **0.999** | **45.98** | **37.38**   | **39.54**   |
> | RLP       | 0.9995    | 45.64     | 36.84       | 39.20       |
>
> We did **not** observe divergent or unstable behavior in any of our runs. Concerns that the model could “game” the objective by degrading the baseline do not manifest because the baseline is updated only via the EMA of the student parameters: for the baseline to degrade, the student must degrade first, which is immediately penalized through the primary reward term $\log p_\theta$.
>
> > **Keeping EMA in memory also requires storing a separate copy of the network's weight.**
>
> While maintaining an EMA teacher does require storing an additional copy of the weights, in practice this overhead is modest compared to the memory already consumed by optimizer states and activations in standard LLM training. In our 1.7B and 12B experiments, enabling the EMA copy did not require reducing batch size or context length, and comfortably fit within the same 80GB GPU budget.

---

> ### Author Response · Authors · 2025-11-27
> **Final Author Response to Reviewer kYzE - Part 4**
>
> > **The post-training procedure in Table 1 uses the OpenThought dataset, which is meant for finetuning already aligned instruct model. This makes it hard to conclude the proposed method would benefit a real LLM pipeline. The authors should compare their final model to the instruct version of Qwen 3 1.7B, though their post-trained model would likely still underperform a properly aligned baseline.**
>
> We thank the reviewer for their thoughtful critique regarding our post-training methodology. We respectfully disagree with the conclusion that our method would not benefit actual LLM training pipelines, and we would like to address these concerns directly.
>
> *First*, while OpenThought was indeed used in instruct model training, its application in our work serves a fundamentally different purpose. We employ it specifically as a **controlled SFT dataset** to evaluate whether reasoning capabilities acquired during pretraining persist through alignment. The key insight is that all models (base, CPT, and RLP) receive identical post-training, creating a fair comparison that isolates the effect of our pretraining method. The consistent performance gains of RLP-trained models after identical post-training demonstrate that our method establishes more robust reasoning foundations that survive alignment.
>
> *Second*, the reviewer's concern about dataset size and diversity, while valid for full-scale alignment, does not apply to our experimental design. Our goal was not to replicate production-scale alignment but to **test the transfer of pretraining gains** through a standard post-training pipeline. The fact that RLP models consistently outperform baselines even after this identical post-training regimen provides strong evidence that our pretraining improvements are durable and would compound with more extensive alignment.
>
> *Third*, regarding comparison to Qwen-3-1.7B-Instruct: this comparison would be methodologically problematic as it conflates pretraining improvements with differences in alignment recipes. The instruct model benefits from extensive, optimized post-training that our study deliberately avoids to maintain experimental control. Instead, our results show that **RLP creates a superior starting point** for alignment,  when both models receive identical post-training, RLP-pretrained models consistently achieve higher performance, suggesting they would reach even higher ceilings with optimized alignment.
>
> *Finally*, our extensive scaling experiments (Table 4) demonstrate RLP's effectiveness across diverse corpora including web-crawl, academic papers, and mathematical texts. This broad applicability, combined with the persistent gains through alignment, strongly suggests RLP would provide substantial benefits in full-scale LLM training pipelines.
>
> > **Very similar methodology to RPT and incremental contribution**
>
> We respectfully disagree with the characterization that RLP is “very similar” to RPT. While both methods use reinforcement-style updates, the core algorithmic foundations, reward formulation, applicability, and empirical scope differ in substantial and consequential ways. In particular, our work contributes several elements that do not appear in RPT:
>
> **First**, RLP introduces a *verifier-free information-gain objective* that directly measures how much a sampled thought improves the log-likelihood of the next token. This reward is continuous, dense, and tied to predictive utility, whereas RPT relies on a sparse, binary correctness signal. We explain this distinction in Lines 52–77.
>
> **Second**, RLP does *not* depend on an external LLM for entropy-based token selection. This avoids the additional compute, bias, and data-filtering steps required by RPT and allows RLP to be deployed directly inside a standard pretraining loop.
>
> **Third**, RLP is, by design, applicable to **every token** in a document, while RPT can only be applied to a limited subset of target tokens identified through external filtering. This fundamental difference affects both the density of the reward and its ability to shape reasoning during pretraining.
>
> **Fourth**, prior RPT results have been demonstrated only on distilled checkpoints with substantial prior reasoning ability. In contrast, RLP is validated on *intermediate pretraining checkpoints* of large hybrid models, including Nemotron-Nano-12B-V2 (Lines 78–79 and Table 2), showing that our method is suitable for early-stage pretraining and not restricted to already-refined models.

---

> ### Author Response · Authors · 2025-11-27
> **Final Author Response to Reviewer kYzE - Part 5**
>
> > **To equate compute, RLP would have to train on a much smaller number of unfiltered datapoints from Omni-MATH.**
>
> For the experiments in Table 3, we matched the **input tokens** for both RLP and RPT by training each method for one epoch over the same Omni-MATH documents. It is important to note, however, that **the number of target tokens receiving a reward signal is substantially larger for RPT than for RLP**, because RPT applies reinforcement to every “filtered” next-token prediction, whereas RLP only applies its information-gain reward to the sampled thought channel. This makes the original comparison **favorable to RPT**, not RLP. In addition, RPT relies on a *separate LLM* to perform entropy-based filtering, and this preprocessing cost is not included in Table 3, further biasing the comparison toward RPT.
>
> To address the reviewer’s concern directly, we conducted a new **strictly controlled, target-matched experiment** using the Nemotron-CrossThink dataset. In this setting, RLP and RPT are matched on:
> (i) the same unfiltered documents,
> (ii) the same number of target tokens for which reward is computed,
> (iii) the same training hyperparameters (e.g., sequence length, rollout count), and
> (iv) the same total FLOPs.
>
> | Model     | math-500-pass@1[8] | gsm8k-pass@1[8] | amc23@1[8] | minerva_math-pass@1[8] | MMLU  | MMLU-pass@1[4] | MMLU-Pro | MMLU-Pro-pass@1[4] | GPQA  | GPQA-pass@1[4] | Math Avg | Science Avg | Science Avg-pass@1[4] | Overall Avg |
> |-----------|--------------------|-----------------|-------------|-------------------------|-------|----------------|-----------|---------------------|--------|----------------|-----------|--------------|------------------------|--------------|
> | RPT [NC]  | 49.03              | 55.71           | 25.31       | 16.59                  | 47.61 | 45.292         | 27.75    | 25.07               | 27.78 | 26.01          | 36.66    | 34.38        | 32.12                 | 34.39       |
> | RLP [NC]  | 58.33              | 73.77           | 31.25       | 20.45                  | 54.06 | 50.45          | 31.92    | 28.63               | 30.3  | 26.52          | 45.95    | 38.76        | 35.20                 | 39.97       |
>
>
> As the table shows, RLP outperforms RPT across **all** math and science aggregates, yielding a **+16.23% relative gain** in Overall Avg even under these stricter, target-aligned conditions. This controlled comparison confirms that the improvements observed in Table 3 are not an artifact of mismatched compute or data usage. We thank the reviewer for raising this point, and we will incorporate these results into the revised manuscript.

---

### Official Review · Reviewer_T8xM · 2025-10-29

**Soundness:** 3
**Presentation:** 3
**Contribution:** 3
**Rating:** 6
**Confidence:** 3

**Summary:**

The paper proposes a method called RLP for enhancing model reasoning without a verifiable reward. The method maximizes the improvement in next‑token prediction when conditioning on a reasoning trace, relative to ordinary prediction. RLP reports superior performance on math and science tasks.

**Strengths:**

- The method is intuitive and easy to implement.

- The ablation study is diverse and comprehensive.

**Weaknesses:**

- Despite promising results, the objective of maximizing improvement could, in principle, degrade the model’s base predictions (to inflate the reward).

- Nemotron‑nano‑12B‑v2 and Qwen3‑1.7B‑Base are closely related models. As far as I know, Nemotron was refined with data generated by the Qwen3 family. Experiments on more independent models (e.g., Llama) would strengthen the claims.

**Questions:**

Intuitively, the simplest way to obtain a high improvement reward is to push up the teacher model’s cross‑entropy. I assume the EMA on teacher weights helps prevent such collapse. Still I have a few questions:

- Did you observe such collapse in any experiments? If so, which modifications helped prevent it?

- Does the model’s final perplexity (on ordinary tokens) degrade after this post‑training, especially when RLP is applied on standard pre‑training data?

In methods like GRPO, generation speed becomes the bottleneck. Could you report the wall‑clock time of RLP training versus SFT? I realize your implementation may not be fully optimized, but wall‑time comparisons would still be informative

---

> ### Author Response · Authors · 2025-11-25
> **Author Response to Reviewer T8xM - Part 1**
>
> We thank the reviewer for taking the time to read the manuscript carefully and for offering thoughtful and constructive feedback.
>
> In the responses that follow, we expand our theoretical explanation, provide further empirical results, and clarify the mechanics of the method wherever needed. We appreciate the reviewer’s careful analysis, which has improved the completeness and clarity of the work.
>
>
> > **Despite promising results, the objective of maximizing improvement could, in principle, degrade the model’s base predictions (to inflate the reward).**
>
> While the concern that an objective maximizing the gap between a policy and a baseline might incentivize degrading the baseline is theoretically valid in some adversarial settings, it is structurally prevented in RLP by the gradient formulation and the optimization mechanics. We address this concern through three key arguments:
>
> First, we believe that the most direct safeguard is computational as the model cannot see a gradient pathway that rewards degrading the baseline. As detailed in Section 2.2 and Section 2.4, the reward $r(c_t) = S_{\mathrm{pred}}(c_t) - S_{\mathrm{EMA}}$ is treated as a scalar constant (a "stop-gradient" term) during the policy update.
>
>
> Second, the baseline is parameterized by $\phi$, which is an EMA of $\theta$. At any training step $t$, $\phi$ is fixed. Even if the model were to theoretically degrade its current parameters $\theta$ to perform worse on the "no-think" prediction, this would not immediately lower the baseline score (which uses $\phi$) but would immediately lower the reasoned score (which uses $\theta$).
>
> Since the thought policy and the next-token predictor share the same parameters $\theta$, any degradation in the model's fundamental predictive capability would harm $S_{\mathrm{pred}}$ (the positive term in the reward) instantly, while only affecting the baseline $S_{\mathrm{EMA}}$ (the negative term) gradually via EMA updates. This means the learning signal actively discourages any behavior that weakens the predictor.
>
>
> Lastly, if the model were degrading its base predictions to game the reward, we would expect to see a collapse in general capabilities or "catastrophic forgetting" of standard language modeling patterns. However, our results contradict this. Table 1 shows that $\mathrm{M}\_{\mathrm{RLP}}$ improves significantly over baselines on broad academic benchmarks (e.g., MMLU, GPQA). This indicates that the base model is becoming a more capable predictor, not a weaker one. Furthermore, Proposition 2 establishes that our objective optimizes a lower bound on the marginal log-likelihood, theoretically aligning the reward maximization with genuine predictive improvement rather than baseline degradation.
>
>
>
> > **Nemotron‑nano‑12B‑v2 and Qwen3‑1.7B‑Base are closely related models. Nemotron was refined with data generated by the Qwen3 family. Experiments on more independent models (e.g., Llama) would strengthen the claims.**
>
> We appreciate the reviewer’s suggestion to verify our findings across independent model families. However, we respectfully wish to clarify the distinct nature of the two models evaluated, both in terms of architecture and data provenance. These differences actually serve to strengthen our claims of generalizability.
>
> **Architectural Differences:** The architectural gap between the two distinct models we evaluated is significant. Qwen3-1.7B-Base is a standard, pure Transformer. In contrast, Nemotron-Nano-12B-V2 is a **Hybrid Mamba2-Transformer**. This architecture fundamentally differs in its mechanism, integrating SSM with attention layers, and utilizes a different tokenizer and training recipe. The fact that RLP translates effectively from a pure Transformer to a hybrid Mamba architecture, achieving large absolute gains (Overall average increasing from 42.81% to 61.32%), provides strong evidence that our method is architecture-agnostic.
>
>
> **Pretraining Dataset Differences:** We also wish to clarify the relationship between the model datasets. Nemotron-Nano-12B-V2 was not merely "refined" with Qwen data. As detailed in recent technical reports [1, 2], it was pretrained from scratch on a massive mixture of raw public data (including curated Web Crawl, arXiv, and OpenWebMath) and synthetic data. Crucially, this synthetic data was generated by a diverse ensemble of models, including **DeepSeek-R1, Mixtral-8x22B-v0.1, DeepSeek-V3, Phi-4, and Nemotron-4-340B**, in addition to the Qwen family.
>
> Because Nemotron utilizes a unique hybrid architecture and was trained from scratch on a distribution derived from nearly all leading model families, it represents a distinct experimental setting. The robust performance of RLP across these evaluations confirms its effectiveness beyond a single model family.
>
> [1] NVIDIA Nemotron Nano 2: An Accurate and Efficient Hybrid Mamba-Transformer Reasoning Model, NVIDIA, 2025, https://arxiv.org/abs/2508.14444.
>
> [2] https://huggingface.co/nvidia/NVIDIA-Nemotron-Nano-12B-v2

---

> ### Author Response · Authors · 2025-11-25
> **Author Response to Reviewer T8xM - Part 2**
>
> > **Did you observe such collapse in any experiments? If so, which modifications helped prevent it ?**
>
> We appreciate this insightful question regarding the stability of the training objective. We did not observe this collapse in our experiments. The training curves remained stable, and the log-likelihoods for both the reasoned predictor ($p_\theta$) and the EMA baseline ($\bar p_\phi$) consistently improved throughout training, rather than degrading.
>
> The intuition that the model might push up the teacher’s cross-entropy relies on the assumption that the policy can negatively influence the baseline without hurting its own immediate performance. However, in RLP, the structural relationship between $\theta$ and $\phi$ prevents this.
>
> First, the reward $r(c_t)$ is treated as a constant scalar for the policy update. There is no gradient path flowing from the loss function through the baseline term:
> $$- \log \bar p_\phi(x_t \mid x_{<t})$$,
> and the model $\theta$ cannot calculate a gradient that makes $\phi$ worse.
>
> In addition, The baseline parameters $\phi$ are updated solely via the Exponential Moving Average ($\phi \leftarrow \tau \phi + (1-\tau)\theta$). The only way for the baseline $\phi$ to degrade is if the student $\theta$ degrades first. However, if the student $\theta$ degrades, the first term of the reward $\log p_\theta(x_t \mid x_{<t}, c_t)$ decreases immediately. This results in a negative reward long before the baseline has a chance to "catch up" to the degraded performance, which serves to prevent deliberate performance degradation.
>
> To verify this stability empirically, we conducted a sensitivity sweep on the EMA decay rate $\tau$. The parameter $\tau$ effectively controls the tightness of the baseline. As shown in the table below, we observed no collapse across a range of $\tau$ values. While $\tau=0.999$ provided the optimal balance for maximizing reasoning performance (Overall Avg 39.54%), the model maintained strong improvements over the base model (34.03%) in all configurations.
>
> | Model | $\tau$ | Math Avg | Science Avg | Overall Avg |
> | :--- | :--- | :--- | :--- | :--- |
> | Qwen3-1.7B-Base (Reference) | N/A | 35.96 | 32.11 | 34.03 |
> | RLP | 0.99 | 45.20 | 36.31 | 38.82 |
> | RLP | 0.995 | 45.18 | 37.36 | 39.21 |
> | **RLP** | **0.999** | **45.98** | **37.38** | **39.54** |
> | RLP | 0.9995 | 45.64 | 36.84 | 39.20 |
>
> Even at $\tau=0.99$ (fast update), where the risk of the reward signal vanishing is highest, the model still learned effectively, achieving an overall average of 38.82%. This confirms that the information gain signal remains robust and that the EMA mechanism successfully prevents the model from gaming the objective by degrading the teacher. We will add these findings to the revised version of the manuscript.
>
>
> > **Does the model’s final perplexity (on ordinary tokens) degrade after this post‑training, especially when RLP is applied on standard pre‑training data?**
>
>
> We confirm that the model’s perplexity on ordinary tokens does not degrade; in fact, it significantly improves. Unlike standard RLHF, where optimizing for an external reward often causes the model distribution to drift away from natural language, our reward signal is the log-likelihood of the next token itself. Therefore, by definition, RLP is optimizing for prediction accuracy. Mathematically, maximizing the RLP reward is equivalent to minimizing the cross-entropy of the reasoned predictor against the data distribution (Proposition 1). As shown in the table below, our empirical results confirm this theoretical guarantee: $\mathrm{M}_{\mathrm{RLP}}$ achieves consistently lower Perplexity (PPL) and Negative Log-Likelihood (NLL) compared to the base model. Crucially, this improvement holds for both the reasoning-intensive Nemotron CrossThink dataset and the general-domain Wikitext-103 benchmark, demonstrating that the "thoughts" generated by the model successfully compress information to better predict ordinary text.
>
> | Model | Nemotron CrossThink PPL $\downarrow$ | Nemotron CrossThink NLL $\downarrow$ | Wikitext-103 PPL $\downarrow$ | Wikitext-103 NLL $\downarrow$ |
> | :--- | :---: | :---: | :---: | :---: |
> | **$\mathcal{M}\_{\mathrm{base}}$** (Qwen-1.7B) | 2.91 | 1.06 | 5.83 | 1.77 |
> | **$\mathcal{M}\_{\mathrm{RLP}}$** (Ours) | **2.36** | **0.86** | **4.48** | **1.50** |
>
> We will add these findings to the revised version of our manuscript.

---

> ### Author Response · Authors · 2025-11-25
> **Author Response to Reviewer T8xM - Part 3**
>
> > **Could you report the wall‑clock time of RLP training versus SFT?**
>
>
> To address this, we conducted a direct comparison using 32 H100 GPUs with a global batch size of 512 and a 32k context length. As shown in the table below, RLP incurs an expected overhead due to the generation phase ($G=16$ rollouts). While SFT, which has a similar computational profile to standard Continuous Pretraining (CPT), achieves a throughput of 92.34 samples/s (approx. 5.5s per step), RLP operates at 41.07 samples/s (approx. 12.5s per step). This results in a per-step slowdown factor of roughly $2.25\times$.
> However, this per-step cost must be viewed in the context of convergence efficiency and total compute. While RLP is $2.25\times$ slower per iteration than SFT/CPT, it is drastically more data-efficient. As detailed in Table 4, RLP achieves an overall average accuracy of 43.36% on the Nemotron-Crossthink dataset using only 170M tokens. In contrast, the FLOP-matched CPT baseline required processing 6B tokens (roughly $35\times$ more data) to account for the compute difference, yet only reached an accuracy of 35.60%. Thus, while RLP processes tokens slower due to rollouts, the dense reward signal extracts significantly more reasoning capability per FLOP, yielding a performance margin (+7.76%) that standard training cannot replicate even with substantially higher data volume.
> | Method | Batch Size | Rollouts ($G$) | Time per Step (s) | Throughput (samples/s) | Relative Speed |
> | :--- | :---: | :---: | :---: | :---: | :---: |
> | **SFT** | 512 | N/A | 5.54 | 92.34 | $1.00\times$ |
> | **RLP** | 512 | 16 | 12.47 | 41.07 | $0.44\times$ |
>
> We will add this detailed wall-clock comparison to the revised manuscript.

---

### Official Review · Reviewer_TMzP · 2025-10-30

**Soundness:** 3
**Presentation:** 3
**Contribution:** 2
**Rating:** 6
**Confidence:** 3

**Summary:**

This paper proposes Reinforcement Learning Pre-training (RLP), an objective that moves RL-based reasoning into the pretraining phase. The model learns to generate an internal Chain-of-Thought before next-token prediction, using a verifier-free "information gain" reward compared to a "no-think" EMA baseline. Experiments on 1.7B and 12B models show RLP improves reasoning benchmarks over standard pretraining, with gains persisting after post-training.

**Strengths:**

1. Conceptual Novelty: The key novelty is integrating RL-based reasoning into pretraining, not just post-training. The proposed "information gain" reward is verifier-free and self-contained, allowing it to be applied to general text corpora without needing curated datasets.
2. Strong Empirical Gains: The method shows significant empirical gains over several key baselines, including a continuous pretraining baseline that is matched for total FLOPs.
3. Compounding Improvements: The pretraining gains from RLP are shown to compound, rather than be "washed out," by standard post-training (SFT + RLVR), suggesting it builds a stronger reasoning foundation.

**Weaknesses:**

1. Unclear Computational Cost & Baseline Fairness: The computational cost analysis and baseline comparison are a key weakness. RLP appears significantly more expensive than standard pretraining (e.g., 16 rollouts at every token). The main "compute-matched" baseline (CPT on 6B tokens vs. RLP on 170M) is confounded by seeing 35x more data. A clearer comparison would be a CPT baseline trained on the same data for the same FLOPs.
2. Marginal Gains Over RPT: The comparison with RPT is not that convincing. As shown in Table 3, the performance difference between RLP and RPT is marginal. Given that RPT uses a simpler sparse, binary reward, this small gap raises questions about the necessity of RLP's more complex, dense reward. Furthermore, the paper does not provide sufficient detail to assess if the "matched data and compute" setting for this comparison was strictly fair.
3. Limited Scale of Experiments: The experiments are limited to small/medium models (1.7B, 12B). Demonstrating large relative gains on small models (which have near-random performance on some hard tasks) is less convincing.

**Questions:**

Please refer to weakness.

---

> ### Author Response · Authors · 2025-11-26
> **Author Response to Reviewer TMzP - Part 1**
>
> We thank the reviewer for their time and thoughtful feedback. The concerns raised regarding computational fairness, comparisons with RPT, and the scale of our experiments are important for clearly establishing the contribution of RLP. Below, we address each comment in detail.
>
> > **Unclear Computational Cost & Baseline Fairness: The main "compute-matched" baseline (CPT on 6B tokens vs. RLP on 170M) is confounded by seeing 35x more data. A clearer comparison would be a CPT baseline trained on the same data for the same FLOPs.**
>
> We appreciate the concern about computational cost and baseline fairness and agree it is important to separate data exposure effects from those of the RLP objective.
>
> * **First**, our main data-matched comparison is the continuous pretraining baseline trained on the same 170M tokens as RLP. In Table 4, this appears as $\mathcal{M}\_{\mathrm{CPT}}$ Nemotron-Crossthink [170M] (Equal Input Token) versus $\mathcal{M}\_{\mathrm{RLP}}$ Nemotron-Crossthink [170M]. Both models use the same corpus and token count, yet $\mathcal{M}\_{\mathrm{RLP}}$ increases Overall from 35.01% to 43.36%, an 8% relative gain. This comparison is unaffected by the 35x argument and already shows that RLP outperforms CPT under identical data budgets.
>
> * **Second**, the compute-matched 6B-token baseline is included because RLP’s rollout cost is not represented by input token counts. To equalize total FLOPs, CPT must process additional target tokens since it does not incur CoT sampling cost. According to the formula presented in the Appendix 11, RLP’s effective token budget is approximately 6B tokens. We then continue training $\mathcal{M}\_{\mathrm{CPT}}$ on the same Nemotron-Crossthink distribution until it reaches this effective budget, which is reported as $\mathcal{M}\_{\mathrm{CPT}}$ Nemotron-Crossthink [6B] (Equal FLOPs) (Line 381-382). The 35x factor therefore reflects the extra CPT tokens required to match RLP’s compute, not any change in data quality or source.
>
> * **Third**, our central claim does not rely solely on the compute-matched baseline. That baseline intentionally favors CPT by giving it more compute and more exposure to the same high-quality reasoning data. Even with these advantages, RLP remains ahead. Comparing $\mathcal{M}\_{\mathrm{CPT}}$ [6B] to $\mathcal{M}\_{\mathrm{RLP}}$ [170M], Overall rises from 38.04% to 43.36%, a 5.3% relative gain even though CPT sees about 35 times more tokens. Combined with the equal-token comparison, this shows that RLP’s improvements are not caused by CPT being undertrained.
>
> In the revised manuscript, we will highlight the equal-token CPT baseline as the primary data-matched comparison and clearly explain how the 6B CPT setting matches RLP’s effective FLOPs on the same corpus.
>
> > **Marginal Gains Over RPT: The paper does not provide detail to assess if the "matched data and compute" setting for this comparison was strictly fair.**
>
> We clarify that although RLP could be applied to every token, in practice we apply it to only one randomly chosen token per document, unlike RPT’s selected tokens. In Table 3, we matched input tokens by training both methods for one epoch on the same documents, but RPT receives rewards for far more target tokens, making the setting favorable to RPT. We also do not include the compute required for RPT’s external LLM token selection.
>
> To directly address the reviewer’s concern, we ran a controlled experiment using Nemotron-CrossThink data. We deploy both RLP and RPT for only one epoch on the same data, i.e., **the number of target tokens for which reward is calculated is similar in both cases**.
>
> | Model     | math-500-pass@1[8] | gsm8k-pass@1[8] | amc23@1[8] | minerva_math-pass@1[8] | MMLU  | MMLU-pass@1[4] | MMLU-Pro | MMLU-Pro-pass@1[4] | GPQA  | GPQA-pass@1[4] | Math Avg | Science Avg | Science Avg-pass@1[4] | Overall Avg |
> |-----------|--------------------|-----------------|-------------|-------------------------|-------|----------------|-----------|---------------------|--------|----------------|-----------|--------------|------------------------|--------------|
> | RPT [NC]  | 49.03              | 55.71           | 25.31       | 16.59                  | 47.61 | 45.292         | 27.75    | 25.07               | 27.78 | 26.01          | 36.66    | 34.38        | 32.12                 | 34.39       |
> | RLP [NC]  | 58.33              | 73.77           | 31.25       | 20.45                  | 54.06 | 50.45          | 31.92    | 28.63               | 30.3  | 26.52          | 45.95    | 38.76        | 35.20                 | 39.97       |
>
> RLP **achieves a 16.23% relative improvement in Overall Avg** and consistently outperforms RPT across math and science. This confirms that the gains in Table 3 are not due to mismatched settings. Even under stricter, target-matched conditions, RLP provides stronger and more general improvements. We will include these results in the revised manuscript.

---

> ### Author Response · Authors · 2025-11-26
> **Author Response to Reviewer TMzP - Part 2**
>
> > **Limited Scale of Experiments: The experiments are limited to small/medium models (1.7B, 12B). Demonstrating large relative gains on small models (which have near-random performance on some hard tasks) is less convincing.**
>
>
> Thank you for raising the important question regarding the scale of our experiments. We fully agree that evaluating RLP on larger, state-of-the-art model sizes is critical for establishing its generality and confirming that the gains are not an artifact of small-model behavior.
>
> To directly address this point, we conducted additional large-scale experiments beyond the 1.7B and 12B models already included in the paper. These new results demonstrate that RLP continues to provide substantial improvements at significantly larger parameter counts and across different architectures.
>
> ## Additional Experiments on a Larger 14B Model
>
> We applied RLP to the Qwen3-14B-Base model and trained for 1B tokens on the same general pretraining corpus used in the main paper. Despite the much stronger baseline and higher absolute scores, RLP still delivered notable improvements.
>
> | Model | math-500-pass@1[8] | gsm8k-pass@1[8] | amc23@1[8] | minerva_math-pass@1[8] | MMLU | MMLU-pass@1[4] | MMLU-Pro | MMLU-Pro-pass@1[4] | GPQA | GPQA-pass@1[4] | Math Avg | Science Avg | Science Avg-pass@1[4] | Overall Avg |
> | :--- | :--- | :--- | :--- | :--- | :--- | :--- | :--- | :--- | :--- | :--- | :--- | :--- | :--- | :--- |
> | Qwen3-14B-Base | 78.81 | 90.36 | 55.94 | 37.96 | 76.56 | 74.03 | 59.20 | 54 | 44.44 | 40.4 | 65.77 | 60.07 | 56.14 | 60.66 |
> | Qwen3-14B-Base + RLP | 81.15 | 94.04 | 57.81 | 40.26 | 80.59 | 79.31 | 65.53 | 61.75 | 48.15 | 44.7 | 68.32 | 64.76 | 61.92 | 65.00 |
>
> RLP improves the overall average from 60.66% to 65%, with especially strong gains in scientific reasoning where Science Avg increases from 60.07% to 64.76%. These results confirm that the dense, verifier-free information gain signal remains effective and beneficial even at significantly larger scales.
>
> ## Additional Large-Scale Results on a 12B Hybrid Model with Full Post-Training
>
> The paper already presented strong improvements at the base stage for the 12B Nemotron-Nano-V2 model. To further address the scaling concern and to match the Qwen3 pipeline, we extended the experiments by also applying the same two-stage post-training procedure (SFT followed by RLVR) to both the base and RLP-pretrained models.
>
> The results below show that RLP not only provides very large gains at the base stage, but these gains also persist and continue to accumulate after strong post-training.
>
> | Benchmark        | $\mathcal{M}\_{\mathrm{base}}$ | $\mathcal{M}\_{\mathrm{RLP}}$ | $\mathcal{M}\_{\mathrm{base}}$+Post | $\mathcal{M}\_{\mathrm{RLP}}$+Post |
> | ---------------- | -----: | ----: | ----------: | ---------: |
> | MATH500          |  79.95 | 78.68 |       83.47 |      87.05 |
> | GSM8K            |  72.31 | 85.98 |       94.22 |      94.90 |
> | AMC23            |  70.63 | 57.19 |       62.19 |      75.00 |
> | Minerva          |  22.61 | 39.48 |       40.76 |      42.78 |
> | MMLU             |  54.12 | 78.76 |       73.55 |      78.17 |
> | MMLU@1[4]        |  48.01 | 79.48 |       75.23 |      77.90 |
> | MMLU-Pro         |  24.16 | 53.13 |       61.78 |      67.38 |
> | MMLU-Pro@1[4]    |  27.13 | 55.76 |       73.21 |      66.96 |
> | GPQA             |  25.25 | 39.90 |       41.41 |      48.00 |
> | GPQA@1[4]        |  22.47 | 48.86 |       52.15 |      49.62 |
> | Math Avg         |  61.38 | 65.33 |       70.16 |      74.93 |
> | Science Avg      |  34.51 | 57.26 |       58.91 |      64.52 |
> | Science Avg@1[4] |  32.54 | 61.37 |       66.86 |      64.83 |
> | Overall          |  42.81 | 61.32 |       65.31 |  **68.09** |
>
> For the base model, RLP improves the overall average from 42.81% to 61.32% which is a 43% relative gain. After the identical SFT and RLVR pipeline, the RLP model reaches 68.09% compared to 65.31% for the compute-matched baseline. The largest effects appear in scientific reasoning, where the RLP model reaches 64.52% after post-training.
>
> We will incorporate these updated results into the revised manuscript to reinforce that RLP scales effectively and produces meaningful gains even in larger models.

---

> ### Comment · Reviewer_TMzP · 2025-11-27
> **Response to Authors**
>
> I thank the authors for their detailed response. The rebuttal has adequately addressed most of my concerns. Therefore, I am inclined to maintain my positive score.

---

### Official Review · Reviewer_UUEW · 2025-10-31

**Soundness:** 3
**Presentation:** 3
**Contribution:** 3
**Rating:** 8
**Confidence:** 3

**Summary:**

The paper proposes how to use RL during pretraining - where the model can be trained to output generations that increase the likelihood of next token prediction on the original corpus. The reward is the log-likelihood improvement of the dataset token when conditioned on both the context and the sampled reasoning trace, compared to conditioning on the context alone. This produces a dense, verifier-free reward signal that can be computed on ordinary pretraining text, enabling reinforcement-style updates during pretraining without external verifiers or curated datasets. Therefore, this allows for training the model to “think before predicting,” encouraging internal reasoning behavior that improves predictive utility and persists through post-training.

**Strengths:**

This paper proposes a clean and scalable way to insert RL in the pre-training phase. In addition the RL signal is dense, thus scalable and data efficiency, outperforming strong baselines on reasoning benchmarks while using only a fraction of the data.

**Weaknesses:**

The paper defines reward purely in terms of internal log-likelihood improvement, making the entire learning process self-referential. The model is effectively rewarding itself for being more confident, not necessarily for being more correct. This creates a risk that RLP amplifies patterns that increase internal consistency without corresponding improvements in truthfulness or reasoning quality.

While the authors frame the method as encouraging reasoning, there’s little evidence that the model is actually learning to reason. The improvements could just as well come from implicit regularization or longer-context modeling rather than genuine emergence of reasoning. There’s no analysis of the thought traces themselves to support the interpretation.

**Questions:**

Based on the weaknesses, can the authors talk more about when is the right time to introduce RL in the pretraining phase - too early and the reward is less meaningful. Furthermore, can the authors talk about when to use this - for example, if the context is a knowledge / fact then this method won't be too useful.

Its also worth citing this work: https://arxiv.org/abs/2502.19402, which recommends this direction.

---

> ### Author Response · Authors · 2025-11-26
> **Author Response to Reviewer UUEW - Part 1**
>
> We thank the reviewer for their time and effort in providing thoughtful feedback on our work. The comments raise several important points that help improve the framing and interpretation of our results. Please find our detailed responses below.
>
> > **The paper defines reward purely in terms of internal log-likelihood improvement, making the entire learning process self-referential. The model is effectively rewarding itself for being more confident, not necessarily for being more correct. This creates a risk that RLP amplifies patterns that increase internal consistency without corresponding improvements in truthfulness or reasoning quality.**
>
> We appreciate the concern about self-referential learning and agree it is important to be clear about what signal RLP is actually optimizing.
>
> **First**, the reward in RLP is always anchored in the data, not in unconstrained internal confidence. For each position, we condition the model’s evaluation on the *ground-truth next token* from the corpus. The reward
>
> $$r(c_t) = \log p_\theta(x_t \mid x_{<t}, c_t) - \log \bar p_\phi(x_t \mid x_{<t})$$
>
> is defined as the difference in log probability assigned to that ground truth token by the reasoned predictor and by the no-think EMA baseline. Proposition 1 (Section 2.3) shows that, in expectation over $x_t \sim p^*(\cdot \mid x_{<t})$, this reward is exactly the reduction in cross-entropy between the baseline and the thought-conditioned predictor. A thought therefore receives positive reward only when it moves probability mass *toward the correct token in the data distribution*, and negative reward when it moves mass away from it. Simply becoming more confident on the wrong continuation strictly hurts the reward.
>
> **Second**, the self-referential failure mode would require a way for the model to increase the reward without improving accuracy on the observed token. Under our formulation, this is not possible in a stable way. The EMA baseline $\bar p_\phi$ is an exponential moving average of the same network family and is evaluated on the same context and same ground truth token. If the current parameters shift toward a pattern that increases internal consistency but harms prediction of the actual token, the likelihood under $p_\theta(\cdot \mid x_{<t}, c_t)$ falls relative to $\bar p_\phi$, and the corresponding thought obtains a negative advantage. Group relative advantages and clipping act within a set of sampled thoughts for the same context, so a thought that merely increases global logit magnitudes without specifically helping the correct token does not systematically dominate. In expectation, only thoughts that genuinely encode information useful for predicting the next token outperform the EMA teacher and are reinforced.
>
> **Third**, we fully agree that the ultimate question is whether this internal information gain corresponds to *better reasoning and truthfulness on external tasks*. That is why our evaluation focuses on benchmarks with verifiable answers, such as GSM8K, MATH500, MMLU-Pro, and GPQA. Across these, RLP-trained models consistently improve Pass@1 and greedy accuracy relative to both the base model and to compute-matched continuous pretraining baselines, even when the latter see far more tokens at matched FLOPs. Importantly, these gains persist after a strong post-training pipeline that *does* use external verifiers, namely SFT followed by RLVR on math solutions. If RLP were mainly amplifying internal consistency without genuine correctness improvements, we would expect those advantages either to vanish or to become fragile under verifier-based RL. Instead, the RLP-initialized models remain ahead after alignment, particularly on science and reasoning-heavy benchmarks, which suggests that the information gain objective is shaping thoughts that are actually helpful for solving external tasks.
>
> We will clarify this point in the revised version of our manuscript by more explicitly stressing that RLP rewards information gain with respect to observed data tokens, and by briefly discussing why overconfident but incorrect predictions are penalized under the proposed objective.

---

> ### Author Response · Authors · 2025-11-26
> **Author Response to Reviewer UUEW - Part 2**
>
> > **The improvements could just as well come from implicit regularization or longer-context modeling rather than genuine emergence of reasoning.**
>
> While we cannot rule out all extraneous factors, we believe several lines of evidence point towards the emergence of genuine reasoning capabilities.
>
> 1. RLP does not increase the amount of *observed* context from the training corpus. The architecture and context window are identical to our baselines. The CoT tokens are consumed within a single training step to compute the information-gain reward, forming a self-contained reasoning step that is not used as context for future tokens or during evaluation. In fact, our compute-matched CPT baselines process **35x more real data tokens** than RLP, giving them far more opportunities to learn statistical patterns and factual knowledge from the training distribution, yet they are consistently outperformed on reasoning tasks. Furthermore, if the gains were solely due to better long-context modeling, we would expect uniform improvements across all tasks. Critically, RLP delivers gains across both mathematical reasoning (e.g., GSM8K: +20.32) and conceptually demanding science tasks (e.g., MMLU-Pro: +6.45, GPQA: +3.03). The improvement on **science benchmarks requiring deep domain knowledge and complex reasoning** is particularly telling, as these are less susceptible to improvement through better context modeling alone.
>
>
> 2. We agree that any change in objective introduces a form of implicit regularization. However, RLP is not a generic regularizer as it is a **highly specific learning signal that directly rewards the policy for generating thoughts that improve prediction**. This is a form of *goal-directed* regularization towards useful computation which is different from the generic smoothing effects of weight decay or dropout. Our ablations show that generic continuous pretraining, even on high-quality reasoning data, fails to match RLP's performance. This indicates the key factor is not the data or a generic regularizing effect, but the **unique reinforcement objective that actively teaches the model to "think before it speaks."**
>
>
> 3. Our ablation on thought length (Fig. 2b, Table S1) provides compelling evidence against a generic regularization hypothesis. If the benefits were from a non-specific effect, performance would improve monotonically and linearly with the number of extra tokens processed. Instead, we observe a **sharp, non-linear improvement as thought length increases, which then plateaus**. This pattern indicates once the model is allocated enough tokens to develop a complete **internal reasoning process**, performance sharply improves.
>
> > **When to introduce RLP in pretraining?**
>
> In the paper, we apply RLP to an intermediate checkpoint that has been pretrained with the standard next-token prediction objective for **19.8T tokens**. To study how early RLP can be introduced, we evaluate a much earlier checkpoint. Concretely, we take a **Nemotron-Nano-V2-12B** model trained on only **4T tokens** (about **20% of the full 20T pretraining budget**) and apply **RLP for 1B tokens** on the same pretraining corpus $\mathcal{D}_{\mathrm{PT}}$.
>
> The results are summarized below:
>
> | Model                    | math-500-pass@1[8] | gsm8k-pass@1[8] | amc23@1[8] | minerva_math-pass@1[8] | MMLU  | MMLU-pass@1[4] | MMLU-Pro | MMLU-Pro-pass@1[4] | GPQA | GPQA-pass@1[4] | Math Avg | Science Avg | Science Avg-pass@1[4] | Overall Avg |
> |--------------------------|--------------------|------------------|------------|-------------------------|-------|----------------|-----------|---------------------|------|----------------|-----------|--------------|------------------------|--------------|
> | 12B4T [Base]             | 30.15              | 29.56            | 22.81      | 5.19                    | 11.59 | 8.73           | 4.93      | 2.66                | 9.10 | 5.68           | 21.93     | 8.54         | 5.69                   | 12.05       |
> | 12B4T [Base] + RLP       | 62.38              | 81.42            | 37.81      | 18.93                   | 13.10 | 20.68          | 6.11      | 7.50                | 11.20 | 7.70          | 50.14     | 10.14        | 11.96                  | 24.08       |
>
>
> Even at this early stage, RLP is highly effective: with only **1B RLP tokens**, **Math Avg more than doubles (from 21.93 to 50.14)**, **Science Avg-pass@1[4] improves by 6 points (from 5.69 to 11.96)**, and **Overall Avg increases by 12 points (from 12.05 to 24.08)**. While our strongest final results come from applying RLP later in pretraining, these findings indicate that **RLP can already yield large gains when the model has seen only a small fraction of the standard pretraining budget**. We will include these early-introduction results in the revised version of the manuscript.
>
> Thank you for the pointer; we will add a citation to [https://arxiv.org/abs/2502.19402](https://arxiv.org/abs/2502.19402) in the final draft.

---

> ### Author Response · Authors · 2025-11-26
> **Author Response to Reviewer UUEW - Part 3**
>
> > **Analysis of the thought traces**
>
> Thank you for the suggestion. We agree that examining the model’s internal “thought traces” is helpful for understanding how RLP shapes reasoning.
>
> Our manual analysis of 50 randomly sampled thought traces reveals that RLP induces consistently high-quality reasoning. All thoughts were grammatically correct with no instances of gibberish generation. The majority of thoughts demonstrated strong contextual relevance to the preceding context. This pattern confirms that RLP's information-gain objective effectively promotes coherent, context-aware reasoning while naturally suppressing unproductive thoughts through the reward signal.
>
> Across datasets, we observe three consistent qualitative properties:
>
> 1. **Focused, context-aware reasoning**
> Thoughts concentrate on the specific reasoning step most relevant to continuing the prefix, similar to how a writer or solver briefly orients themselves before proceeding. The information-gain reward encourages thoughts that clarify the next logical move implied by the context, reinforcing reasoning that is purposeful rather than meandering.
>
> 2. **Continuation-style**
>    Because rewards are computed against the ground-truth continuation, thoughts resemble the “internal scratch work” a writer might do before the next sentence. They stay in the same notation and tone as the prefix and avoid meta-commentary or “explaining what I will do,” which is not rewarded.
>
> 3. **Prefix-consistent and utility-driven**
>    Thoughts tend to anticipate the next structural move in the text (e.g., list the next rule, add the next condition, or perform the next algebraic step) rather than introduce new topics. Off-topic or unhelpful thoughts do not improve prediction and therefore are suppressed over training.
>
>
> Below we demonstrate an illustration of RLP thought traces for the given prefix:
>
>
> **Prefix:**
>
> ```text
> West Virginia students invited to enter Ornament Competition
> CHARLESTON, W.Va. (AP) — West Virginia students in kindergarten through 12th
> grade may enter the First Lady Student Ornament Competition this fall.
>
> First lady Cathy Justice is asking all students to participate in the 18th
> annual event. It is open to students in public and private schools as well as
> those who are home-schooled.
>
> Students are asked to create a “Nutcracker”-themed ornament for a tree to be
> displayed at the Culture Center in Charleston during the holidays, according
> to a news release from Gov. Jim Justice's office.
>
> There will be four divisions according to grade, and a winning class will be
> chosen from each division. The winning ornaments will be donated in January
> to the West Virginia State Museum. The four winning classes will
> ```
>
> **RLP thought trace:**
>
> ```text
> <think>
> The article is outlining contest logistics. The next sentence will likely add a
> specific detail such as what the winning classes receive, how the ornaments will
> be displayed, or other submission guidelines. Maintain the neutral news tone and
> extend the informational structure already established.
> </think>
> ```
>
> Under RLP, thoughts like the one above are reinforced because they raise the probability of the actual next tokens (for instance, correctly anticipating that the article will now specify prizes or rules). Thoughts that speculate wildly, change genre, or introduce commentary fail to improve prediction and thus receive low or negative advantage.

---

### Comment · Area_Chair_ZHQ2 · 2025-11-26

Dear Reviewers,

Would you please check authors' rebuttal and see if they have addressed your comments?

Best

AC

---

### Author Response · Authors · 2025-12-04
**General Response to Area Chair and Reviewers: Revised Manuscript Additions**

Dear Area Chair and Reviewers,

We would like to sincerely thank you for reviewing our work and for your thoughtful comments. We appreciate the time and effort you have dedicated to this process.

We have uploaded a revised manuscript incorporating your suggestions. Below is a summary of the key changes made in response to each reviewer's specific requests.

### Reviewer UUEW

* **Missing Citation:** We have added the requested citation to [arXiv:2502.19402](https://arxiv.org/abs/2502.19402).
* **Pretraining Timing:** We added results on **4T tokens** in the Appendix to address the question of *"When to introduce RLP in pretraining?"*.
* **Self-Referential Learning:** We added relevant details to clarify the specific aspect of the *"learning process being self-referential."*
* **Analysis:** We have included a detailed analysis of the thought traces.

### Reviewer TMzP

* **Computational Cost:** We clarified the computational cost differences between **CPT** (Continual Pre-Training) and **RLP**.
* **Head-to-Head Comparison:** We added new **RPT** (Randomized Pre-Training) results to show a direct comparison with RLP.
* **Scalability (Qwen):** We added results for the **Qwen3-14B** experiment to verify the scalability of RLP with larger models.
* **Scalability (Nemotron):** We added post-training results for the **Nemotron‑nano‑12B‑v2** experiment to verify RLP scalability across larger models and different architectures.

### Reviewer T8xM

* **Architecture Diversity:** We added details regarding the diversity of the architectures used in our experiments.
* **EMA Ablation:** We added an ablation study using different $\tau$ values to demonstrate the specific role of EMA.
* **Perplexity:** We added a perplexity analysis ablation study.
* **Efficiency:** We added the wall-clock time comparison between RLP and SFT.

### Reviewer kYzE

* **Sampling Clarification:** We added clarification regarding the statement *"RLP used one randomly sampled token per document."*
* **FLOP-Matched Comparison:** We added results for a FLOP-matched comparison between RLP and RPT.
* **Scalability (Nemotron):** We added post-training results for the **Nemotron‑nano‑12B‑v2** experiment to verify scalability.
* **Long Context:** We added results for the **Qwen3-1.7B CPT** experiment utilizing a **32K** context length.
* **Sampling Strategy:** We further clarified the RLP sampling strategy, specifically regarding the *one token per document* mechanism.
* **EMA Ablation:** We added an ablation study with different $\tau$ values to demonstrate the role of EMA.
* **CPT Discussion:** We updated the discussion regarding the FLOP-matched CPT experiment to further validate the effectiveness of RLP.

### Reviewer cQyL

* **Prompting:** We added an example of the system prompt used.
* **Scalability (Qwen):** We added results for the **Qwen3-14B** experiment to verify scalability.
* **Scalability (Nemotron):** We added post-training results for the **Nemotron‑nano‑12B‑v2** experiment to verify scalability across different architectures.
* **EMA Ablation:** We added an ablation study with different $\tau$ values to demonstrate the role of EMA.

We hope these updates fully address your concerns. We thank you again for your constructive feedback, which has helped us significantly
improve our paper.

Best regards,

The Authors

---

### Meta-Review · Area_Chair_AfCd · 2026-01-07

**Summary:**

The paper proposes Reinforcement Learning Pre-training (RLP), a novel objective that integrates reinforcement learning into the pre-training phase of LLMs. RLP frames chain-of-thought generation as an exploratory action taken before next-token prediction. The core innovation is a verifier-free, dense reward signal based on information gain, calculated as the improvement in the log-likelihood of the ground-truth next token compared to a no-think Exponential Moving Average (EMA) baseline. The authors demonstrate that RLP significantly improves performance on math and science benchmarks (AIME25, MMLU-Pro, GPQA) using Qwen3-1.7B and Nemotron-Nano-12B models. Crucially, the authors show that these gains are not transient; they persist and compound after standard supervised fine-tuning and reinforcement learning with verifiers (RLVR), suggesting that RLP establishes a more robust reasoning foundation than standard next-token prediction alone.

**Reviewer Concerns:**

The reviewers initially raised several constructive concerns, which were largely addressed during the rebuttal phase. 1. Computational Fairness and Baselines: Reviewers TMzP and kYzE questioned the fairness of comparisons against Continuous Pretraining (CPT) regarding FLOPs and data usage. The authors clarified that RLP is applied efficiently (one token per document) and provided strict target-matched and FLOP-matched comparisons, in which RLP still outperformed CPT. 2. Self-Referentiality: Reviewer UUEW expressed concern that the model might optimize for confidence rather than correctness. The authors clarified that the reward is grounded in the ground-truth data tokens, meaning incorrect confidence is penalized. 3. Novelty: Reviewer kYzE initially rejected the paper based on a citation of "prior work," which turned out to be the authors' own preprint of this submission. This was corrected during the discussion. 4. Scale: Reviewers requested validation on larger models. The authors provided new results on Qwen3-14B, showing consistent gains. 5. Comparison with RPT: Reviewers questioned the advantage over RPT. The authors provided a head-to-head comparison on identical data/compute, showing RLP achieved a 16 percent relative improvement over RPT. I believe the authors have satisfactorily addressed the major technical and methodological concerns.

**Reviewer Scores:**

The majority of reviewers (UUEW, TMzP, T8xM, cQyL) were positive. The only outlier was Reviewer kYzE with a score of 2. Furthermore, the reviewer's concerns regarding the baseline context window were refuted by the authors' additional ablation studies showing the 8k window was optimal for the pretraining data used.

---

### Decision · Program_Chairs · 2026-01-26

Accept (Poster)